# Interferon-epsilon is a novel regulator of NK cell responses in the uterus

Jemma R Mayall [ID][1,8], Jay C Horvat[1,8], Niamh E Mangan[2], Anne Chevalier[1], Huw McCarthy[1], Daniel Hampsey[1], Chantal Donovan[3], Alexandra C Brown[1], Antony Y Matthews[2], Nicole A de Weerd [ID][2], Eveline D de Geus [ID][2], Malcolm R Starkey [ID][1,4], Richard Y Kim[1,3], Katie Daly[1], Bridie J Goggins[1], Simon Keely[1], Steven Maltby [ID][1], Rennay Baldwin[1], Paul S Foster[1], Michael J Boyle [ID][1,5], Pradeep S Tanwar [ID][6], Nicholas D Huntington[7], Paul J Hertzog[2,9] & Philip M Hansbro [ID][1,3,9] ✉

## Abstract

**The uterus is a unique mucosal site where immune responses are balanced to be permissive of a fetus, yet protective against infections. Regulation of natural killer (NK) cell responses in the uterus during infection is critical, yet no studies have identified uterine-specific factors that control NK cell responses in this immune-privileged site. We show that the constitutive expression of IFNε in the uterus plays a crucial role in promoting the accumulation, activation, and IFNγ production of NK cells in uterine tissue during *Chlamydia* infection. Uterine epithelial IFNε primes NK cell responses indirectly by increasing IL-15 production by local immune cells and directly by promoting the accumulation of a pre-pro-like NK cell progenitor population and activation of NK cells in the uterus. These findings demonstrate the unique features of this uterine-specific type I IFN and the mechanisms that underpin its major role in orchestrating innate immune cell protection against uterine infection.**

**Keywords** Interferon-Epsilon; Type I Interferon; Natural Killer Cell; Female Reproductive Tract; *Chlamydia* Infection
**Subject Categories** Immunology; Urogenital System

## Introduction

The interactions between infection and the mucosal immune environment in the female reproductive tract (FRT) are complex, and how many of the factors that restrict infection are regulated, versus those that allow for tolerance of a semi-allogeneic fetus, are yet to be clarified. *Chlamydia trachomatis* frequently causes severe reproductive tract sequelae in women, such as pelvic inflammatory disease (PID), tubal infertility and ectopic pregnancy. Strong interferon (IFN)γ and adaptive Th1 responses promote the clearance of *Chlamydia* and limit the development of disease (Cohen et al, 2005; Cotter et al, 1997; Debattista et al, 2002; Johansson et al, 1997; Kaiko et al, 2008; Perry et al, 1997; Wang et al, 1999) and the production of IFNγ during the early stages also plays an important role in controlling *Chlamydia* infections. Previous studies showed that NK cells are the predominant source of IFNγ early during *Chlamydia* infections, with NK cell depletion resulting in diminished IFNγ expression, a shift from Th1-dominant to Th2-dominant responses, and delayed clearance (Tseng and Rank, 1998).

The expression of cytokines and activating receptors by both host epithelial and accessory immune cells recruit NK cells and promote their cytokine production and cytolytic activity. Significantly, type I IFNs play important roles in regulating NK cell responses by direct and indirect mechanisms, and protective NK cell responses are dependent on type I IFN signaling during other intracellular infections, however, the particular IFN responsible has not been characterized (Martinez et al, 2008; Zhu et al, 2008).

We previously showed that the novel type I IFN, IFNε, is constitutively and most highly expressed in the uterus by endometrial epithelial cells, is under hormonal regulation, and protects against *Chlamydia* and herpes simplex virus (HSV)-2 infections from the earliest stages (Fung et al, 2013; Stifter et al, 2018). This indicates that IFNε is critical in potentiating protective innate immune responses in the FRT, however, how it affects immune responses and the mechanisms by which it mediates defense against infection are not yet known.

In this study, we show that IFNε is responsible for promoting the infiltration, activation, and IFNγ production of NK cells in the uterus during *Chlamydia* infection. IFNε primes for these

[1]Immune Health Program, Hunter Medical Research Institute and the University of Newcastle, Newcastle, NSW 2308, Australia. [2]Centre for Innate Immunity and Infectious Diseases, Hudson Institute of Medical Research and Departments of Molecular and Translational Sciences, Monash University, Clayton, VIC 3168, Australia. [3]Centre for Inflammation, Centenary Institute and University of Technology Sydney, Faculty of Science, School of Life Sciences, Sydney, NSW 2000, Australia. [4]Immunology and Pathology, Central Clinical School, Monash University, Clayton, VIC 3168, Australia. [5]Immunology and Infectious Diseases Unit, John Hunter Hospital, Newcastle, NSW 2305, Australia. [6]Gynecology Oncology Research Group, School of Biomedical Sciences and Pharmacy, University of Newcastle, Newcastle, NSW 2308, Australia. [7]Monash Biomedicine Discovery Institute, Monash University, Clayton, VIC 3168, Australia. [8]These authors contributed equally: Jemma R Mayall, Jay C Horvat. [9]These authors contributed equally as senior authors: Paul J Hertzog, Philip M Hansbro. ✉E-mail: philip.hansbro@uts.edu.au

protective NK cell responses by increasing IL-15 production, promoting the accumulation of a pre-pro-like NK cell progenitor population, and directly activating NK cells in the uterus. These IFNε-mediated effects on NK cells are important in protecting against infection. These findings represent new functions for IFNε in driving protective immunity in this unique mucosal site.

# Results

## IFNε deficiency results in decreases in NK cell responses in the uterus

We previously showed that *Ifne*$^{-/-}$ mice have increased *Chlamydia* load from 3 days post-infection (dpi) in a mouse model of FRT infection (Fung et al, 2013). To determine if IFNε is involved in the recruitment of immune cells to the FRT during the early stages of infection, we performed flow cytometry to quantify common leukocyte populations (Hickey et al, 2011) in uterine tissues from *Chlamydia*-infected WT and *Ifne*$^{-/-}$ mice. *Ifne*$^{-/-}$ mice have 42.7% fewer conventional (c)NK cells (CD45$^+$CD3$^-$NK1.1$^+$) in uterine tissues compared to WT mice during infection at 3dpi (mean total number of NK cells: IFNε$^{-/-}$ mice [$9.46 \times 10^3$], WT controls [$1.65 \times 10^4$]; Fig. 1A). cNK cells are also the most abundant immune cell in the FRT at this time point of infection (Fig. EV1A). There are no significant differences in macrophages, plasmacytoid DCs, myeloid DCs, CD4$^+$ T cells, CD8$^+$ T cells, B cells, or NK T cells and a small reduction in neutrophils in infected *Ifne*$^{-/-}$ compared to WT mice. We also assessed the effects on innate lymphoid cells (ILC1-3: CD45$^+$ Lin$^-$ IL-7Rα$^-$ CD90.2$^+$ T-bet$^{+/-}$) in the uterus, however, they were absent, or too few present to examine, during *Chlamydia* infection (Fig. EV1B). The changes in NK cell numbers early during infection are associated with an increase in gross oviduct pathology, measured by oviduct cross-sectional area, in *Ifne*$^{-/-}$ mice at 14dpi (Fig. EV1C). Increases in oviduct size are indicative of the development of hydrosalpinx, one of the key features of *Chlamydia*-induced pathology in both humans and mice (Lee et al, 2020). Since previous studies also showed 3dpi to be the peak of NK cell infiltration in this model of *Chlamydia* infection (Tseng and Rank, 1998), we focused on this time point to further characterize the effects of IFNε on NK cell responses.

We used intracellular cytokine staining and flow cytometry to determine the effects of IFNε deficiency on NK cell number, phenotype, activation, and IFNγ production during *Chlamydia* infection in the FRT following the gating strategy shown in Fig. EV1D–F. *Ifne*$^{-/-}$ mice not only had fewer cNK cells (Fig. 1A), but also fewer tissue-resident uterine (u)NK cells (CD45$^+$CD3$^-$NK1.1$^-$CD49b$^-$CD122$^+$; Fig. 1B) during infection compared to WT mice. However, outside of pregnancy, uNK cells constitute a small proportion of the NK cell population in the FRT (here they were 4.32% of NK cells). The following analyses focused on cNK cells and all NK cells are conventional unless otherwise specified. Populations of activated CD69$^+$ and IFNγ$^+$ and IFNγ$^-$ NK cells are also altered in the absence of IFNε (Figs. 1C–G and EV2A–F). *Ifne*$^{-/-}$ mice have fewer CD69$^+$IFNγ$^+$ double-positive (Fig. 1C,D) and CD69$^+$IFNγ$^-$ (Fig. 1C,E) single-positive NK cells in the upper FRT during infection compared to WT controls. Curiously, however, there are no differences in the numbers of

CD69$^-$IFNγ$^+$ single-positive (Fig. 1C,F) and CD69$^-$IFNγ$^-$ double-negative NK cells (Fig. 1C,G). Additionally, the proportion of the NK cell population expressing CD69 and/or IFNγ is reduced while the proportion negative for both factors is increased in infected *Ifne*$^{-/-}$ compared to WT mice (Figs. 1C and EV2F). This demonstrates that the reductions in the numbers of active and IFNγ-producing NK cells in the FRT are not solely due to a decrease in the total numbers of NK cells present, but are also due to a shift in the frequency of activation and cytokine production. In WT mice, NK cells represented ~60% of all IFNγ-producing leukocytes during infection and the decreases in IFNγ-producing NK cells in *Ifne*$^{-/-}$ mice correspond with reductions in the total numbers of IFNγ-producing leukocytes (Fig. EV2G). Importantly, the reductions in IFNγ$^+$ cells in infected *Ifne*$^{-/-}$ mice correlate with decreases in IFNγ protein production in uterine lavage measured by ELISA (Fig. 1H).

Since NK cells can influence T cell polarity and T cells are another source of IFNγ during *Chlamydia* infection, we also assessed the effect of IFNε on IFNγ production by T cells during infection. However, very few T cells are present in the upper FRT and the numbers of IFNγ-producing T cells are unaltered in *Ifne*$^{-/-}$ mice at this time-point of infection (Fig. EV2H).

## IFNε deficiency results in decreases in systemic NK cells during FRT infection

We next examined whether these IFNε-mediated effects on NK cell number, activity, and IFNγ production are specific to the FRT or are also systemic, both at baseline and during infection. *Ifne*$^{-/-}$ mice have similar numbers of splenic NK cells at baseline, but fewer total (Fig. 1I) and activated (Fig. EV2I) splenic NK cells during *Chlamydia* infection, compared to WT controls. However, the proportions of CD69$^{+/-}$IFNγ$^{+/-}$ splenic NK cells are similar between WT and *Ifne*$^{-/-}$ mice, both at baseline and during infection (Figs. 1J and EV2J–M). This indicates that the decreases in activated splenic NK cells in *Ifne*$^{-/-}$ mice are due to overall reductions in total numbers of NK cells rather than a change in the frequency of their activation. No differences are observed in numbers of splenic IFNγ$^+$ NK cells between *Ifne*$^{-/-}$ and WT mice (Fig. EV2N).

cNK cells develop and mature in the bone marrow before entering the circulation. Immature NK cells express NK1.1 but do not acquire CD49b and CD11b until they reach functional maturity (Fathman et al, 2011; Kim et al, 2002; Rosmaraki et al, 2001; Williams et al, 2000). We show that, while there are no differences in the numbers of immature NK cells (CD45$^+$Lin$^-$CD122$^+$NK1.1$^+$CD11b$^-$) in bone marrow between *Ifne*$^{-/-}$ and WT mice (Fig. EV2O), there is a significant reduction in the numbers of CD11b$^+$ mature NK cells (CD45$^+$CD3$^-$CD122$^+$NK1.1$^+$CD11b$^+$) in bone marrow from infected *Ifne*$^{-/-}$ compared to WT mice (Fig. 1K).

## IFNε deficiency reduces numbers of pre-pro NK cell progenitor-like cells in the uterus

Given that IFNε deficiency results in decreases in mature NK cell numbers systemically and locally in the infected FRT, we next determined if NK cell progenitors are also affected. The bone marrow is the primary site of NK cell haematopoiesis, however, early NK cell progenitors also reside in secondary lymphoid tissues and migrate to the uterus and other organs where they proliferate and differentiate

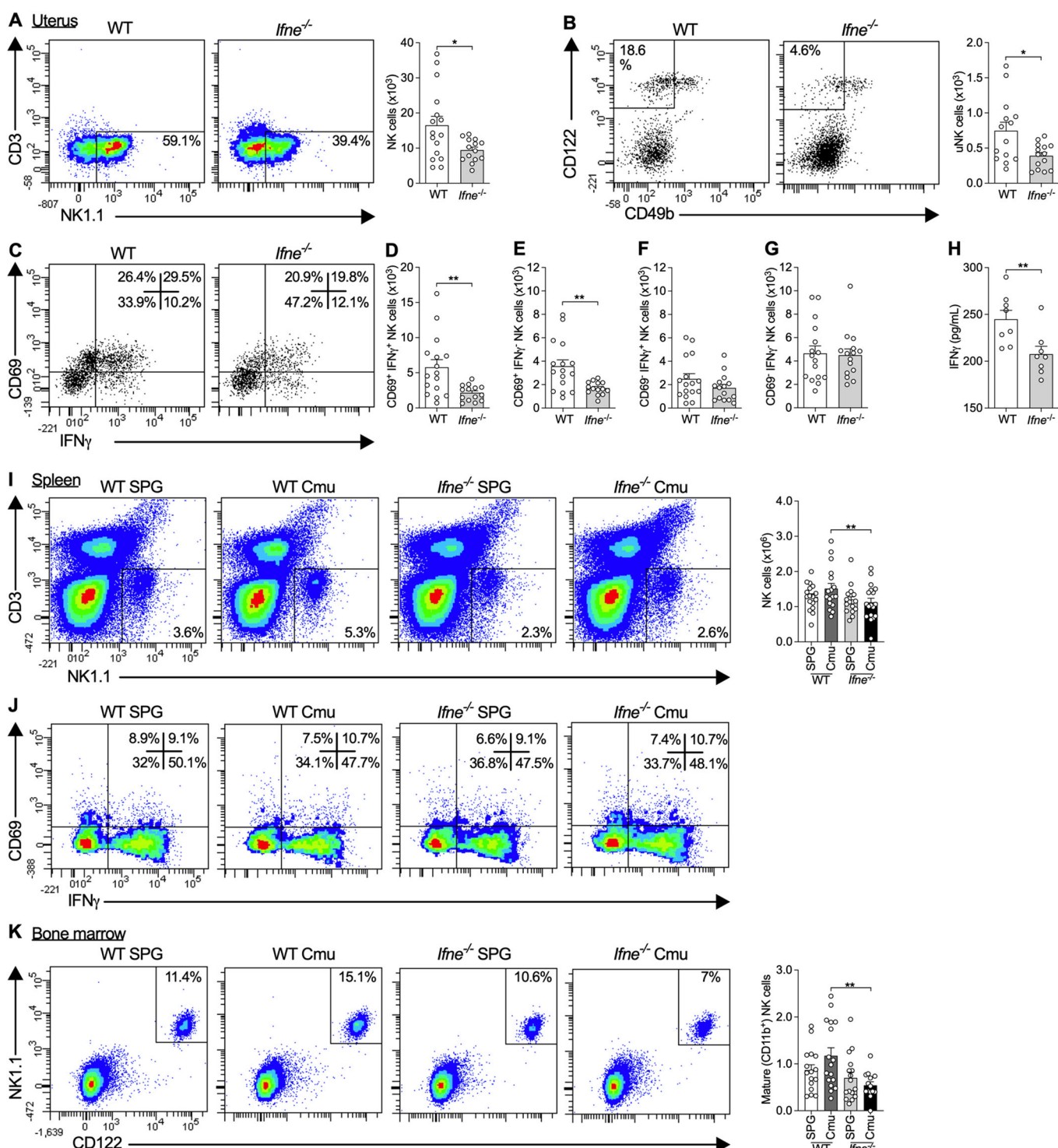

(Chantakru et al, 2002; Male et al, 2010; Vacca et al, 2011). Thus, we assessed the numbers of two NK cell progenitor populations in the bone marrow, uterus, lymph nodes and spleens of *Chlamydia*- and sham-infected *Ifne*−/− and WT mice. Pre-pro NK cells are the earliest committed progenitors of NK cells (Fathman et al, 2011).

In bone marrow, numbers of murine pre-pro NK cells (CD45+Lin−FLT3−IL-7Rα+C-kit^low/−CD122−NK1.1−CD49b−NKG2D+Sca-1+ (Yu et al, 2013)) are similar between *Ifne*−/− and WT mice (Fig. 2A).

However, using bone marrow as a positive control and to set gating for these progenitor populations, we also identify a small population of pre-pro NK cells in the uterus (Fig. 2B). These uterine pre-pro NK cells are rare in sham-infected controls but accumulate during early infection to a more substantial population and are reduced in both *Chlamydia*-and sham-infected *Ifne*−/− compared to WT mice. *Ifne*−/− and WT mice have similar proportions of pre-pro-like NK cells present in the lymph nodes (lumbar aortic and medial iliac lymph nodes) and spleen (Fig. 2C,D).

◀

**Figure 1. Interferon (IFN)ε deficiency results in decreases in the numbers, activation, and IFNγ production of conventional (c) and tissue-resident uterine (u) natural killer (NK) cells in the uterus and decreases in the numbers, but not activation of, NK cells in the spleen and bone marrow during *Chlamydia* infection.**

(A) Flow cytometry of uterine horn cells from *Ifne*[-/-] and wild-type (WT) C57BL/6 mice on day 3 of *Chlamydia muridarum* infection, showing cNK cells (FSC[low-int] SSC[low] CD45[+] CD3[-] NK1.1[+]) and quantification. (B) Flow cytometry of uterine horn cells as in (A), showing uNK cells (FSC[low-int] SSC[low] CD45[+] CD3[-] NK1.1[-] CD49b[-] CD122[+]) and quantification. (C–G) (C) Flow cytometry of uterine horn cells as in (A), showing CD69[+/-] IFNγ[+/-] cNK cells and quantification of (D) CD69[+] IFNγ[+], (E) CD69[+] IFNγ[-], (F) CD69[-] IFNγ[+], and (G) CD69[-] IFNγ[-] cNK cells in uterine horns. (H) Concentrations of IFNγ in uterine lavage fluid. (I) Flow cytometry of splenocytes from *Ifne*[-/-] and WT C57BL/6 mice on day 3 of *Chlamydia muridarum* (Cmu) or sham (SPG) infection, showing conventional NK cells (FSC[low-int] SSC[low] CD45[+] CD3[-] NK1.1[+]) and quantification. (J) Flow cytometry of splenocytes as in (I), showing CD69[+/-] IFNγ[+/-] conventional NK cells. (K) Flow cytometry of bone marrow from femurs of mice as in (I), showing mature conventional NK cells (FSC[low-int] SSC[low] CD45[+] lin[-] CD11b[+] CD122[+] NK1.1[+]) and quantification. Data information: The % displayed on the flow cytometry plots are the % of the parent population the cells within the gates/quadrants comprise. All data presented as mean ± SEM, with individual values. *p < 0.05, **p < 0.01 ((A, B, D–H): two-tailed Mann–Whitney test; (I–K): one-way ANOVA). (A–G): n ≥ 15 (data are from three experiments), (H): n = 8 (data from one experiment), (I–K): n ≥ 15 (data from two experiments; all biological replicates). For flow cytometry on uterine tissue, uteri from 2 to 4 mice were pooled for each biological replicate and at least 15 pooled samples were analyzed. See also Figs. EV1 and EV2. (B) is repeated in Fig. EV1F. Source data are available online for this figure.

The next stage of NK cell development is differentiation into an IL-15 receptor-expressing precursor NK cell by acquisition of the IL-2 receptor β chain (CD122) (Rosmaraki et al, 2001). We identify precursor NK cells (CD45[+]Lin[-]FLT3[-]IL-7Rα[+]C-kit[low/-]CD122[+]NK1.1[-]CD49b[-]NKG2D[+] (Yu et al, 2013)) in the bone marrow, lymph nodes and spleen but not the uterus and numbers were not significantly altered between *Ifne*[-/-] and WT mice (Fig. 2E–G). Together, these data suggest that IFNε is priming a pool of NK progenitors locally that could give rise to protective NKs.

## IFNε deficiency results in a reduction in IL-15 levels in the uterus

To elucidate how IFNε primes for local and systemic NK cell responses, we then assessed expression levels of factors involved in NK cell development, chemoattraction, activation, and IFNγ responses in uterine tissue from *Chlamydia*- and sham-infected, *Ifne*[-/-] and WT mice. *Il15* transcript expression is reduced in *Ifne*[-/-] mice (~40%) at baseline (p = 0.056) and during infection (p = 0.076) compared to WT mice (Fig. 3A). Similarly, IL-15 protein levels are reduced (~25%) in uterine lavage from *Ifne*[-/-] compared to WT mice, particularly during infection (p < 0.05; Fig. 3B). *Cxcl10*, *Il12b* and *Il18* expression levels are unaltered between *Ifne*[-/-] and WT mice (Fig. EV3A–C).

## IFNε expression is linked to NK cell responses in human uterine tissue

We then determined if the link between IFNε, and IL-15 and NK cell responses is also present in humans. mRNA expression levels of *IFNE*, *IL15*, and *NCR1* (NKp46), an activating NK cell receptor whose expression is indicative of NK cell responses in tissue (Ascierto et al, 2013), were assessed in biopsies of healthy uterine tissue from women, and correlation analyses performed between these factors. Both *IL15* (Fig. 3C) and *NCR1* (Fig. 3D) levels positively correlate with *IFNE* expression. *NCR1* levels also positively correlate with *IL15* levels (Fig. 3E).

## IFNε and IL-15 are expressed by different cells

Then, to determine whether IFNε and IL-15 are expressed by the same cells, we performed immunofluorescence staining on uterine tissue sections of IL-15-CFP reporter mice. As expected, IFNε occurs in luminal and glandular epithelial cells (Fig. 3F; arrowhead) but no IL-15 is detected in these cells. Instead, IL-15 expressing cells are detected in uterine stroma (Fig. 3F; filled arrows).

Surprisingly, a subpopulation of stromal cells expressed IFNε and some of these cells also co-expressed IL-15 (Fig. 3F; open arrow).

We then performed flow cytometry to further characterize these cells (Fig. EV3D–G). As expected, almost all epithelial cells expressed IFNε (Fig. 3G), but not IL-15 (Fig. 3H). Notably, a large population of CD45[+] leukocytes expressed IL-15 (Fig. 3H) while only a small subset of leukocytes expressed IFNε (Fig. 3G). Further analysis of the IL-15[+] CD45[+] population revealed that these cells did not express NK cell (NKp46), B cell (CD19; Fig. 3I), or T cell (CD4, CD8; Fig. 3J) markers but were primarily CD11b[+] CD11c[neg-low] MHC-II[-] Ly6G[-] cells (Fig. 3K,L), a surface marker expression pattern indicative of monocytes/macrophages. To determine the extent to which IL-15 is co-expressed with IFNε, we then characterized the expression of both cytokines in common immune cell subsets in the uterus. In uninfected uteri, NK cells are a minor immune cell population (~1–2% of CD45[+] cells). Small proportions of the immune cell subsets assessed co-expressed IFNε and IL-15, with 14.2% of NK cells, 2% of B cells, 2% of CD4[+] cells, 3.2% of CD8[+] cells and 5.5% of monocytes/macrophages expressing both cytokines (Fig. EV3I–K).

Collectively, these data show that the majority of IFNε and IL-15 are produced by distinct populations of cells. Epithelial cells are the main producers of IFNε but do not express IL-15. Most IL-15 is produced by CD45[+] CD11b[+] CD11c[neg-low] MHC-II[-] Ly6G[-] cells. However, a minor subset of uterine immune cells expresses both IFNε and IL-15 which could also contribute to the regulation of NK cell accumulation and function.

## IFNε and IL-15 have independent and synergistic effects on NK cell function

We next determined if IFNε has direct effects on NK cell function and delineated these from IL-15-dependent effects. Circulating CD49b[+] NK cells were isolated from the spleen of naïve mice and stimulated with recombinant (r)IFNε, rIL-15, or a combination, with/without neutralizing anti-IL-15 antibody. Activation and cytokine production upon PMA/ionomycin stimulation (needed in order to detect cytokines in these cells by flow cytometry), and cytolytic activity against YAC-1 target cells was assessed at 18 hours of cytokine stimulation, a timepoint previously shown to be appropriate for the assessment of these responses (Fehniger et al, 2007; Keppel et al, 2015). rIFNε has no effect while rIL-15 increases the numbers of NK cells in overnight cultures (Fig. 4A). Stimulation with either rIFNε or rIL-15 increases the proportion of CD69[+]IFNγ[+] NK cells (2.81- and 3.44-fold, respectively), whereas stimulation with both cytokines in combination increases

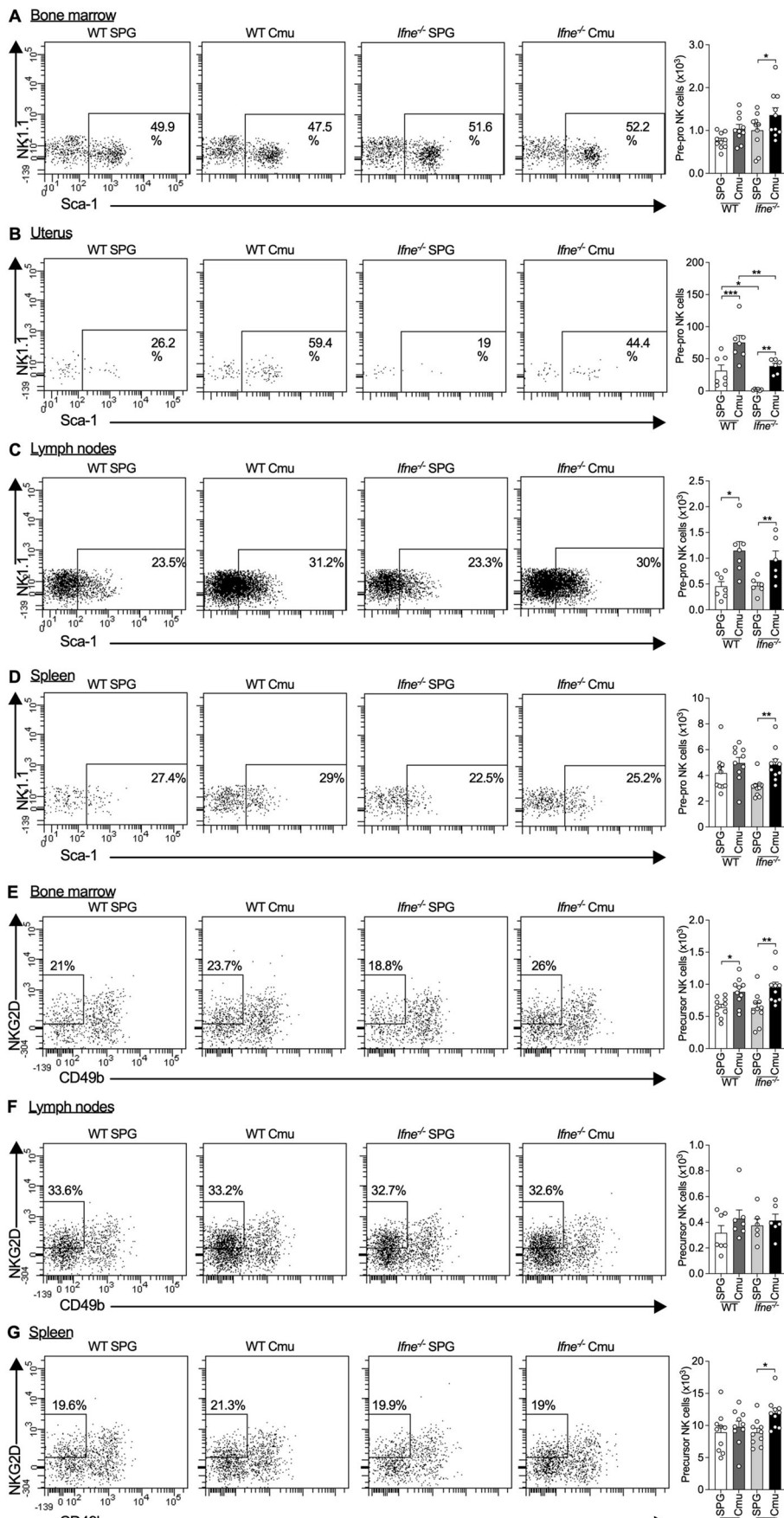

**Figure 2.    Interferon (IFN)ε deficiency results in decreases in the numbers of pre-pro natural killer (NK) cell progenitors in the uterus but not systemically in the bone marrow, lymph nodes, or spleen.**

(A) Flow cytometry of bone marrow from femurs of wild-type (WT) and *Ifne*^-/-^ C57BL/6 mice on day 3 of *Chlamydia muridarum* (Cmu) or sham (SPG) infection, showing pre-pro NK cell progenitors (FSC^low-int^ SSC^low^ CD45^+^ lin^-^ FLT3^-^ IL-7Rα^+^ C-kit^low/-^ CD122^-^ NK1.1^-^ CD49b^-^ NKG2D^+^ Sca-1^+^) and quantification. (B) Flow cytometry of uterine horn cells from mice as in (A), showing pre-pro NK cells and quantification. (C) Flow cytometry of lymph node cells from mice as in (A), showing pre-pro NK cells and quantification. (D) Flow cytometry of splenocytes from mice as in (A), showing pre-pro NK cells and quantification. (E) Flow cytometry of bone marrow from femurs of *Ifne*^-/-^ and WT C57BL/6 mice on day 3 of *Chlamydia muridarum* (Cmu) or sham (SPG) infection showing precursor NK cells (FSC^low-int^ SSC^low^ CD45^+^ lin^-^ FLT3^-^ IL-7Rα^+^ C-kit^low/-^ CD122^+^ NK1.1^-^ CD49b^-^ NKG2D^+^) and quantification. (F) Flow cytometry of lymph node cells as in (E), showing precursor NK cells and quantification. (G) Flow cytometry of splenocytes as in (E) showing precursor NK cells and quantification. Data information: The % displayed on the flow cytometry plots are the % of the parent population the cells within the gates comprise. All data presented as mean ± SEM, with individual values. *$p < 0.05$, **$p < 0.01$, ***$p < 0.001$ (one-way ANOVA). $n \geq 6$, all data from one experiment; biological replicates. For flow cytometry on uterine and lymph node cells, uteri or lymph nodes from 4 mice were pooled for each biological replicate and at least 6 pooled samples were analyzed. (A, E) are repeated in Fig. EV1G. Source data are available online for this figure.

this population to a much greater extent (14.23-fold) (Fig. 4B,C). rIFNε alone increases the proportion of CD69^+^IFNγ^-^ NK cells, however rIL-15 alone has no effect on this population and no additive effect occurs with combination (Fig. 4B,D). Conversely, stimulation with rIL-15 increases the proportion of CD69^-^IFNγ^+^ NK cells, whereas rIFNε has no effect and no additive effects are observed (Fig. 4B,E). The proportion of CD69^-^IFNγ^-^ NK cells is reduced by either rIFNε or rIL-15, and combination further reduces this population (Fig. 4B,F). rIFNε does not affect cytolytic activity, however, rIL-15 increases lysis of YAC-1 cells and the combination increases this further (Fig. 4G). Anti-IL-15 antibody was used to inhibit any autocrine IL-15 produced in response to rIFNε, however, the addition of anti-IL-15 to rIFNε-stimulated NK cell cultures has no effect on their function compared to rIFNε alone (Fig. 4A–G). NK cells from *Ifne*^-/-^ mice have reduced cytolytic activity compared to WT controls (Fig. 4H). Stimulating NK cells from *Ifne*^-/-^ mice with rIL-15 restores their cytolytic activity, however, rIFNε stimulation of these cells has no effect. Thus, rIFNε alone directly stimulates NK cell CD69 expression, while rIL-15 is required to increase NK cell number, IFNγ production, and cytolytic activity.

## Local rIFNε administration increases IL-15 and NK cell responses and reduces *Chlamydia* infection

To confirm the role of IFNε in inducing IL-15 and NK cell responses, we then performed gain-of-function studies in WT mice by transcervical and intravaginal administration of rIFNε prior to infection (Fig. EV4A,B). rIFNε-treated mice have significantly more NK cells in the uterus at 3dpi compared to vehicle (PBS)-treated controls (Fig. 5A). Treatment increases CD69^+ and -^ IFNγ^+ and -^ NK cell populations in the uterus during infection (Figs. 5B–F and EV4C,D) but does not increase the numbers of mature NK cells systemicly (Fig. 5G). There is, however, a trend towards increases in the numbers of immature NK cells in the bone marrow (Fig. EV4E). Treatment also increases the numbers of pre-pro-like NK cells in the uterus during infection (Fig. 5H). These changes in NK cell responses are associated with increases in *Il15* expression in the uterus (Fig. 5I) and decreases in vaginal *Chlamydia* load (Fig. 5J) at 3dpi.

## Local rIL-15 administration increases local and systemic NK cell responses and reduces *Chlamydia* infection

We next determined the role of IL-15 in mediating protective NK cell responses downstream of IFNε in vivo. rIL-15 was transcervically and intravaginally administered to WT mice prior to *Chlamydia* infection,

and its effects on NK cell responses and infection assessed. Like with rIFNε, rIL-15 treatment significantly increases NK cells in the uterus at 3dpi, compared to vehicle-treated controls (Fig. 6A). Treatment also increases CD69^+ and -^ IFNγ^+ and -^ NK cell populations during infection, however, there are greater increases in CD69^-^ and IFNγ^-^ populations, as differences in CD69^+^IFNγ^+^, total CD69^+^ (IFNγ^+ or -^) and total IFNγ^+^ (CD69^+ or -^) populations do not quite reach statistical significance ($p > 0.074$; Figs. 6B–F and EV4F,G). Unlike with rIFNε, rIL-15 treatment increases mature NK cell numbers systemically during infection (Fig. 6G). There is also a trend towards increases in immature bone marrow NK cell numbers (Fig. EV4H). Treatment does not significantly affect pre-pro-like NK cell numbers in the uterus during infection (Fig. 6H) and does not change *Ifne* expression (Fig. 6I), demonstrating that IL-15 is incapable of driving IFNε expression and acts downstream. These changes in NK cell responses track with decreases in vaginal *Chlamydia* infection (Fig. 6J).

## Local rIL-15 administration to *Ifne*^-/-^ mice partially restores NK cell numbers but not activation in the uterus

We next determined if restoring IL-15 levels in *Ifne*^-/-^ mice could restore protective NK cell responses during infection. rIL-15 was administered to *Ifne*^-/-^ mice transcervically and intravaginally prior to infection, and NK cell responses and infection assessed. rIL-15-treated *Ifne*^-/-^ mice had no significant increases in total NK cell numbers in the uterus during infection compared to vehicle-treated *Ifne*^-/-^ controls ($p = 0.254$; Fig. 7A). Treatment of *Ifne*^-/-^ mice is also unable to restore CD69^+^ populations of active NK cells (CD69^+^IFNγ^+^, $p = 0.338$; CD69^+^IFNγ-, $p = 0.221$) during infection, however, there are increases in inactive CD69^-^IFNγ^-^ NK cells to similar levels as in WT mice (Figs. 7B–F and EV5A,B). This increase in inactive NK cells is associated with increases in mature (Fig. 7G) and immature (Fig. EV5C) NK cells in the bone marrow. Treatment also substantially reduces *Chlamydia* infection (Fig. 7H).

## Local rIFNε administration to *Il15*^-/-^ mice does not augment protective responses or decrease *Chlamydia* infection

We next compared the effects of IL-15 and NK cell deficiency to those of IFNε deficiency and determined if IL-15 and NK cells are required for IFNε-mediated protection against infection. Due to the importance of IL-15 in NK cell development, *Il15*^-/-^ mice are completely deficient in mature NK cells (Kennedy et al, 2000) and thus, we do not detect NK cells in their FRTs during infection (Fig. EV5D). This absence of NK cells is associated with reduced total

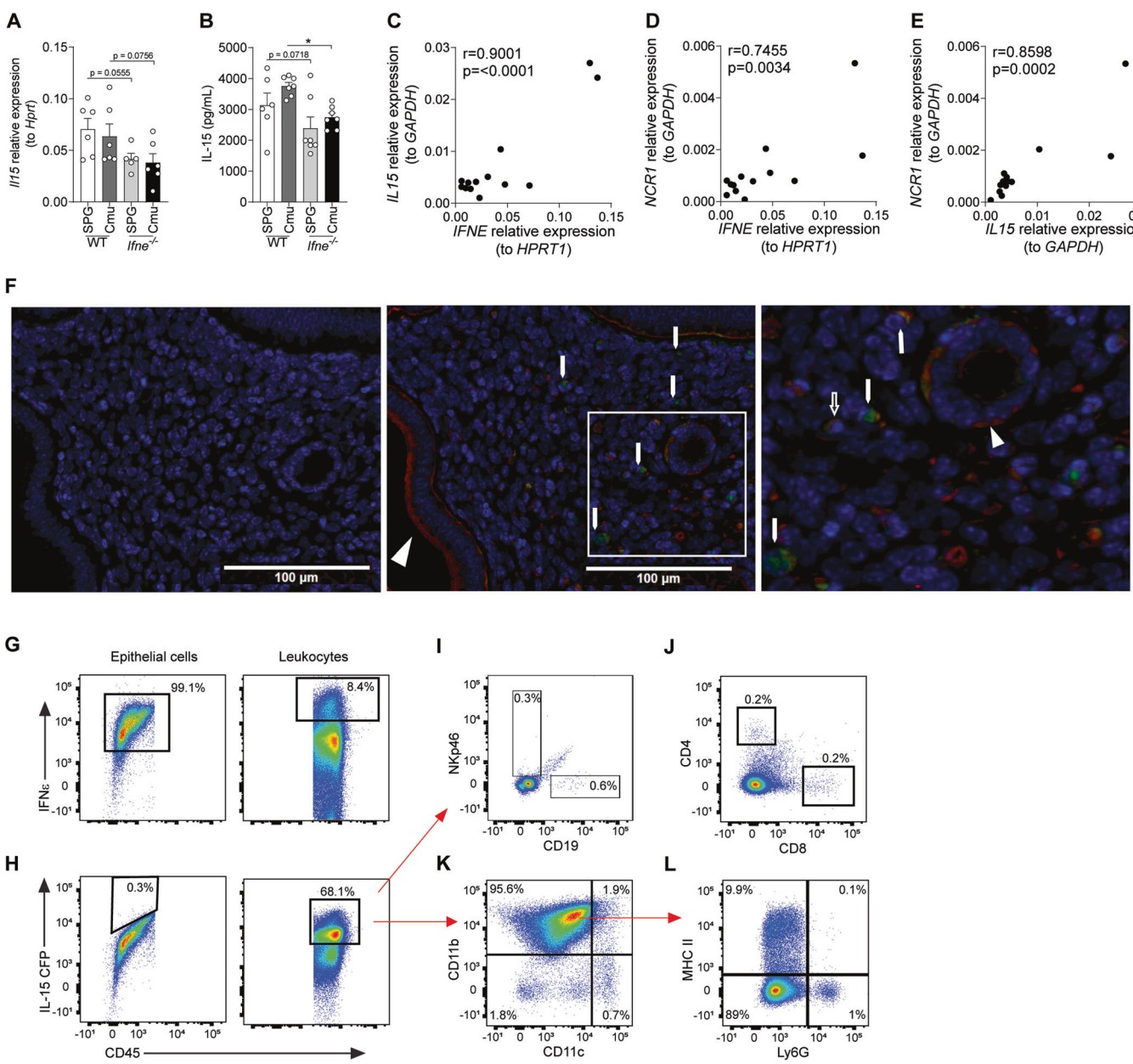

**Figure 3. Interferon (IFN)ε expression correlates with IL-15 production in the murine and human uterus.**

(A) qPCR analysis of *Il15* expression, normalized to the expression of the housekeeping gene *Hprt*, in uterine horns from *Ifne*[-/-] and wild-type (WT) C57BL/6 mice on day 3 of *Chlamydia muridarum* (Cmu) or sham (SPG) infection. (B) Concentrations of IL-15 protein in uterine lavage fluid from mice as in (A). (C–E) qPCR analysis and correlation between (C) *IL15* and *IFNE*, (D) *NCR1* and *IFNE*, and (E) *NCR1* and *IL15* expression, normalized to the expression of the housekeeping genes, *GAPDH* or *HPRT1*, as indicated, in human endometrial tissue samples from 13 donors. (F) Immunofluorescence staining of IFNε (red) and IL-15 (green) in uteri from IL-15-CFP reporter mice. Nuclei are counterstained with DAPI (blue). Left micrograph shows isotype control staining, middle micrograph shows stained section and right micrograph shows a higher magnification of the boxed area indicated in middle micrograph. Arrowheads indicate IFNε immunoreactivity in luminal and glandular epithelial cells, filled arrows indicate IL-15 immunoreactivity in uterine stroma, open arrows indicate IFNε and IL-15 dual immunoreactivity in a subpopulation of stromal cells. (G) Flow cytometry of IFNε expression in epithelial (Pan-Cytokeratin[+]) and immune cells (CD45[+]), respectively, from uteri of IL-15-CFP reporter mice. (H) IL-15 expression in epithelial and immune cells, respectively. (I) Expression of NKp46 and CD19 on IL-15[+] CD45[+] cells. (J) Expression of CD4 and CD8 on IL-15[+] CD45[+] cells. (K) Expression of CD11b and CD11c on IL-15[+] CD45[+] cells. (L) Expression of MHC-II and Ly6G on CD11b[+] CD11c[neg-low] IL-15[+] CD45[+] cells. Data information: The % displayed on the flow cytometry plots are the % of the parent population the cells within the gates/quadrants comprise. Data in (A) and (B) presented as mean ± SEM, with individual values, data in (C–E) presented as individual values. *$p < 0.05$ ((A, B): one-way ANOVA; (C–E): Pearson correlation analysis. (A): $n \geq 5$, (B): $n \geq 7$, (C–E): $n = 13$ (data are from one experiment each; all biological replicates). Immunofluorescence micrographs are representative of 4 different uteri. For flow cytometry, uteri from 3 mice were pooled and 4 pooled samples were analyzed. See also Fig. EV3. (G, H) are repeated in Fig. EV3F,G. Source data are available online for this figure.

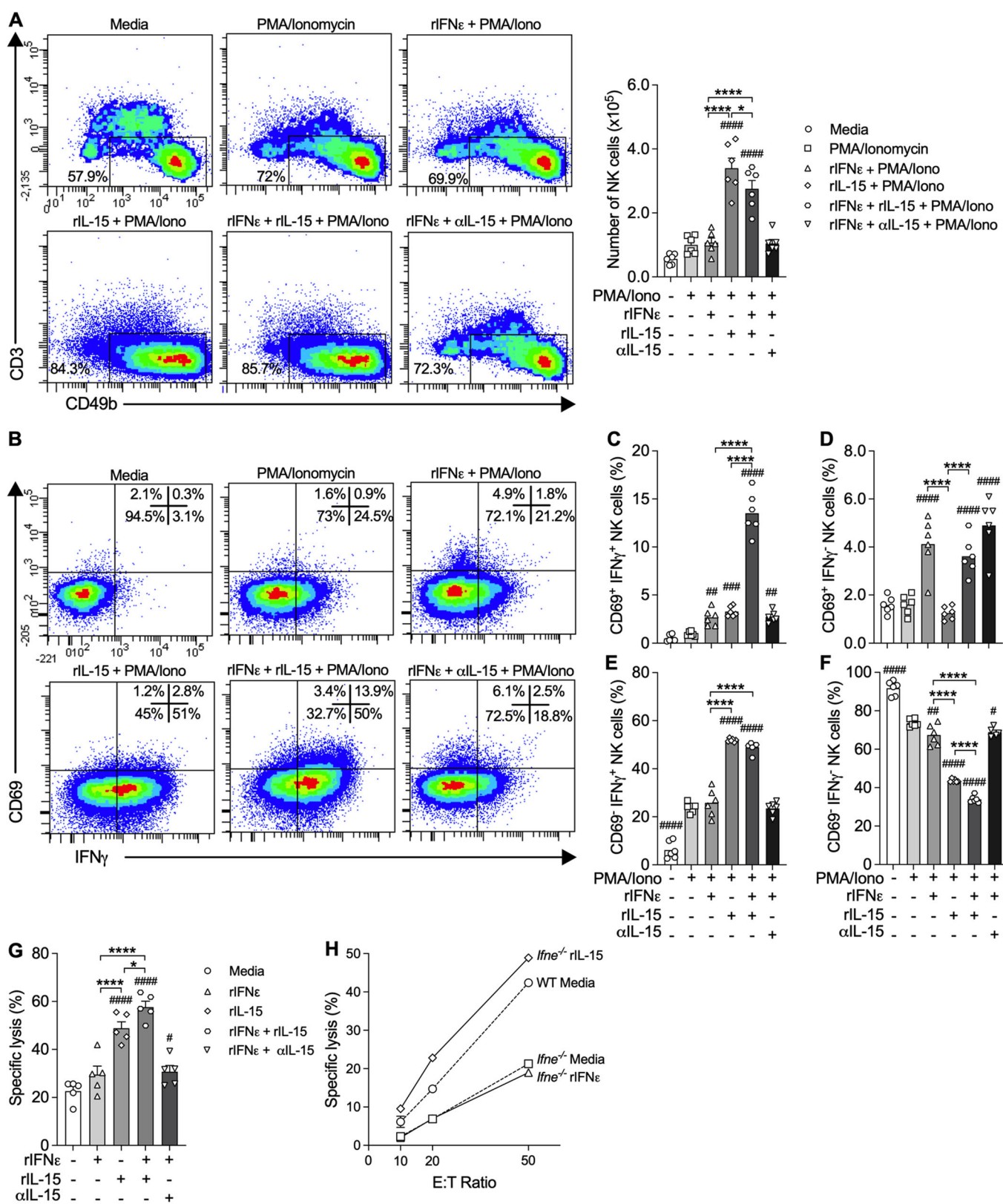

**Figure 4.  Recombinant interferon (rIFN)ε directly and independently increases activation, while rIL-15 is required to increase the number, IFNγ production and cytolytic activity of NK cells in vitro.**

(A) Flow cytometry of conventional NK cells (FSC^low-int SSC^low CD45^+ CD3^- CD49b^+) isolated from the spleens of wild-type (WT) C57BL/6 mice, incubated ex vivo with rIFNε, rIL-15 and/or anti-IL-15 antibody for 13 h then stimulated with PMA and ionomycin with Brefeldin A for a further 5 h and quantification. (B) Flow cytometry of samples as in (A), showing CD69^+/- IFNγ^+/- NK cells. (C–F) Frequency of (C) CD69^+ IFNγ^+, (D) CD69^+ IFNγ^-, (E) CD69^- IFNγ^+ and (F) CD69^- IFNγ^- NK cells in samples as in (B). (G) Percent specific lysis of YAC-1 target cells after 4 h of co-culture with NK cells from the spleens of WT C57BL/6 mice, pre-stimulated with rIFNε, rIL-15 and/or anti-IL-15 antibody for 18 h, at an effector cell:target cell (E:T) ratio of 10:1. (H) Percent specific lysis of YAC-1 target cells after 4 h of co-culture with NK cells isolated from WT and Ifne^-/- C57BL/6 mice, pre-stimulated with rIFNε or rIL-15 for 18 h at the indicated E:T ratios. Data information: The % displayed on the flow cytometry plots are the % of the parent population the cells within the gates/quadrants comprise. All data presented as mean ± SEM. *p < 0.05, ****p < 0.0001; #p < 0.05, ##p < 0.01, ###p < 0.001, ####p < 0.0001 compared to (A, C–F) PMA and ionomycin only control or (G) media only control (A, C–G: one-way ANOVA). (A, C–G): n = 6 (data from one experiment; biological replicates), (H): n ≥ 3 (data from one experiment; technical replicates). Source data are available online for this figure.

numbers of IFNγ-producing leukocytes (Fig. EV5E) and *Ifng* transcript levels in the upper FRT during infection (Fig. 7I). *Il15^-/-* mice have a 24-fold increase in *Chlamydia* burden compared to WT controls (Fig. 7J). WT and NK cell-deficient *Il15^-/-* mice were then treated with rIFNε prior to infection. In WT mice, rIFNε treatment increases *Ifng* transcript levels in the uterus during infection, however, rIFNε treatment does not induce these IFNγ responses in *Il15^-/-* mice (p = 0.527; Fig. 7K). *Il15^-/-* mice have increased *Chlamydia* burden compared to WT controls that is not reduced by rIFNε treatment (p = 0.517; Fig. 7L).

Thus, restoring IL-15 increases the numbers of inactive, but has little effect on CD69^+ NK cells during infection in *Ifne^-/-* mice, indicating that IFNε and its IL-15-independent effects are required for local activation. Similar to *Ifne^-/-* mice, NK cell deficient *Il15^-/-* mice have reduced IFNγ responses during infection and rIFNε is unable to induce IFNγ responses or protect against infection in NK cell-deficient *Il15^-/-* mice. This provides further evidence that NK cells are required to mediate the protective effects of IFNε and that IL-15 mediates some, but not all, of the NK cell mediated effects of IFNε.

## Discussion

Here, we demonstrate that IFNε deficiency leads to a reduction in NK cell numbers that is associated with increased infection and upper RT pathology. We also used NK cell deficient *Il15^-/-* mice and rIFNε to show that NK cells are the major contributors to IFNε-mediated infection control in vivo. This extends our previous studies where we found that bacterial burden is higher throughout *Chlamydia* infection in *Ifne^-/-* mice, from as early as 3dpi and out to 30dpi (Fung et al, 2013). Notably, Tseng and Rank showed that depletion of NK cells is associated with delayed clearance of *Chlamydia* infection, however, their study did not investigate the role of NK cells in protection against ascending infection and pathology in the upper FRT (Tseng and Rank, 1998). We also previously found that IFNε has direct effects on infection in epithelial cells (Stifter et al, 2018). Thus, collectively, these findings show that IFNε has important roles in protecting the FRT from infection and associated pathology, largely by increasing NK cell responses, in addition to directly suppressing infection in epithelial cells.

Previous studies showed that NK cells are important in immune protection against *Chlamydia* infections through the production of IFNγ (Jiao et al, 2011; Tseng and Rank, 1998), subsequent promotion of protective Th1 adaptive immunity *via* modulation of DC function (Jiao et al, 2011), and killing of *Chlamydia*-infected cells (Hook et al, 2004; Ibana et al, 2012). Here we demonstrate, for the first time, that IFNε is an important mediator of these critical immune responses in the uterus. We also show the mechanisms by which IFNε primes for NK cell responses to infection are through driving the accumulation of NK cell progenitors and the expression of IL-15 in the uterus and by directly stimulating NK cell activation. Our data is consistent with reports of conventional type I IFNs (Andoniou et al, 2005; Gill et al, 2011; Lucas et al, 2007) mediating NK cell responses in other tissues, including the production of IFNγ and cytolytic activity. In this study, we show, using *Ifne^-/-* mice and rIFNε, that it is the constitutive, local and hormonally-induced production of IFNε that regulates the accumulation, maturation, cytotoxicity, CD69 expression and effector IFNγ levels of NK cells in the FRT through IL-15-dependent and -independent mechanisms (Athié-Morales et al, 2008; Borrego et al, 1999; Moretta et al, 1991; Jiao et al, 2011).

We demonstrate that the effects of IFNε on NK cell activation and IFNγ-production are restricted to the uterus and do not occur systemically. However, we show that IFNε does increase the numbers of NK cells systemically during infection. The bone marrow is the primary site of NK cell haematopoiesis (Colucci et al, 2003; Haller and Wigzell, 1977), with NK cells mobilizing into the circulation and homing to specific organs or sites of immune responses upon maturation (Mayol et al, 2011; Sciumè et al, 2011). Interestingly, we also identify a population of early NK cell committed progenitors in the uterus and demonstrate that they are regulated by IFNε. These findings are consistent with studies showing that mature NK cells both expand in the periphery upon stimulation (Tsujimoto et al, 2005; Warren, 1996), and early NK progenitors reside in secondary lymphoid tissues and migrate to the uterus and other organs where they proliferate and differentiate (Chantakru et al, 2002; Male et al, 2010; Vacca et al, 2011). Our novel data suggest that IFNε mediates NK cell accumulation in the uterus by not only promoting their bone marrow haematopoiesis and increasing the systemic pool but by uniquely driving their expansion and differentiation locally. This shows a unique role for uterine IFNε regulating NK homeostasis in a non-lymphoid organ.

IFNε promotes NK cell responses in the uterus, in part, *via* the induction of IL-15. This is consistent with reports that other type I IFNs induce IL-15 (Baranek et al, 2012; Lucas et al, 2007). This cytokine is essential for NK cell development and survival (Kennedy et al, 2000; Mrozek et al, 1996; Puzanov et al, 1996; Ranson et al, 2003), and contributes to their recruitment, expansion, activation, and IFNγ production during infection (Allavena et al, 1997; Elpek et al, 2010; Lucas et al, 2007; Nguyen et al, 2002). It also plays a critical role in in situ maturation of uNK

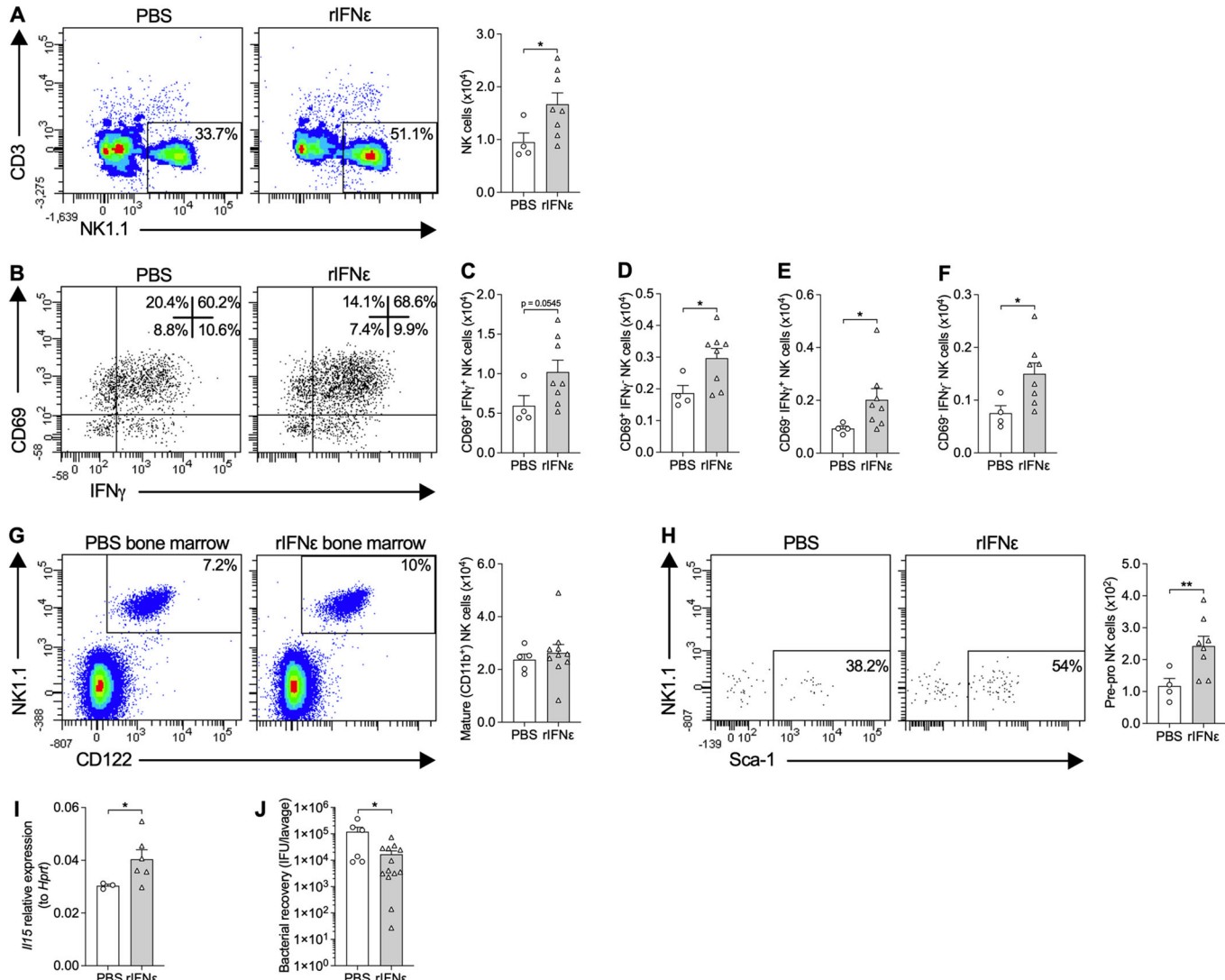

**Figure 5. Local administration of recombinant interferon (rIFN)ε increases NK cells, pre-pro NK cell progenitors and IL-15 expression, and decreases *Chlamydia* infection in the female reproductive tract (FRT).**

(A) Flow cytometry of uterine horn cells from wild-type (WT) C57BL/6 mice prophylactically administered rIFNε or phosphate-buffered saline (PBS) vehicle control transcervically, on day 3 of *Chlamydia muridarum* infection showing conventional NK cells (FSC$^{low-int}$ SSC$^{low}$ CD45$^+$ CD3$^-$ NK1.1$^+$) and quantification. (B) Flow cytometry of uterine horn cells as in (A), showing CD69$^{+/-}$ IFNγ$^{+/-}$ conventional NK cells. (C–F) Quantification of (C) CD69$^+$ IFNγ$^+$, (D) CD69$^+$ IFNγ$^-$, (E) CD69$^-$ IFNγ$^+$ and (F) CD69$^-$ IFNγ$^-$ conventional NK cells in uterine horns as in (B). (G) Flow cytometry of bone marrow from femurs as in (A), showing mature NK cells (FSC$^{low-int}$ SSC$^{low}$ CD45$^+$ lin$^-$ CD11b$^+$ CD122$^+$ NK1.1$^+$) and quantification. (H) Flow cytometry of uterine horn cells as in (A), showing pre-pro NK cell progenitors (FSC$^{low-int}$ SSC$^{low}$ CD45$^+$ lin$^-$ FLT3$^-$ IL-7Rα$^+$ C-kit$^{low/-}$ CD122$^-$ NK1.1$^-$ CD49b$^-$ NKG2D$^+$ Sca-1$^+$) and quantification. (I) qPCR analysis of *Il15* expression, normalized to the expression of the housekeeping gene *Hprt*, in uterine horns as in (A). (J) qPCR analysis of *Chlamydia* numbers normalized to inclusion forming unit (ifu) standards of *Chlamydia* in vaginal lavage fluid from mice as in (A). Data information: The % displayed on the flow cytometry plots are the % of the parent population the cells within the gates comprise. All data presented as mean ± SEM, with individual values. *$p < 0.05$, **$p < 0.01$ ((A, C–J): one-tailed Mann–Whitney test). (A–F, H–I): $n \geq 4$, (G): $n \geq 5$, (J): $n \geq 8$ (all data from one experiment; biological replicates). For flow cytometry on uterine tissue, uteri from 1 to 3 mice were pooled for each biological replicate and at least 4 pooled samples were analyzed. See also Fig. EV4. Source data are available online for this figure.

cells (Allen and Nilsen-Hamilton, 1998; Ye et al, 1996). Our studies on human tissue show a close association between *IFNE* and *IL15* expression. The levels of the NK cell marker, *NCR1*, also positively correlate with both *IFNE* and *IL15* expression in uterine tissue providing evidence of links between IFNε, IL-15, and NK cell responses in the humans. We also show that IL-15 is produced by myeloid cells in response to IFNε signaling from endometrial epithelial cells. The combination of high CD11b, low CD11c and

absence of MHC-II expression indicates that these IL-15-producing cells are not one of the common DC subtypes, while the absence of Ly6G precludes neutrophils (Liu et al, 2020; Merad et al, 2013; Yasuda et al, 2020). This surface marker expression profile is consistent with monocyte/macrophage phenotypes reported across many mouse tissues (Liu et al, 2020). IL-15 is produced by macrophages and stromal cells in the decidua (Gordon, 2021), however, studies characterizing cytokine and surface marker

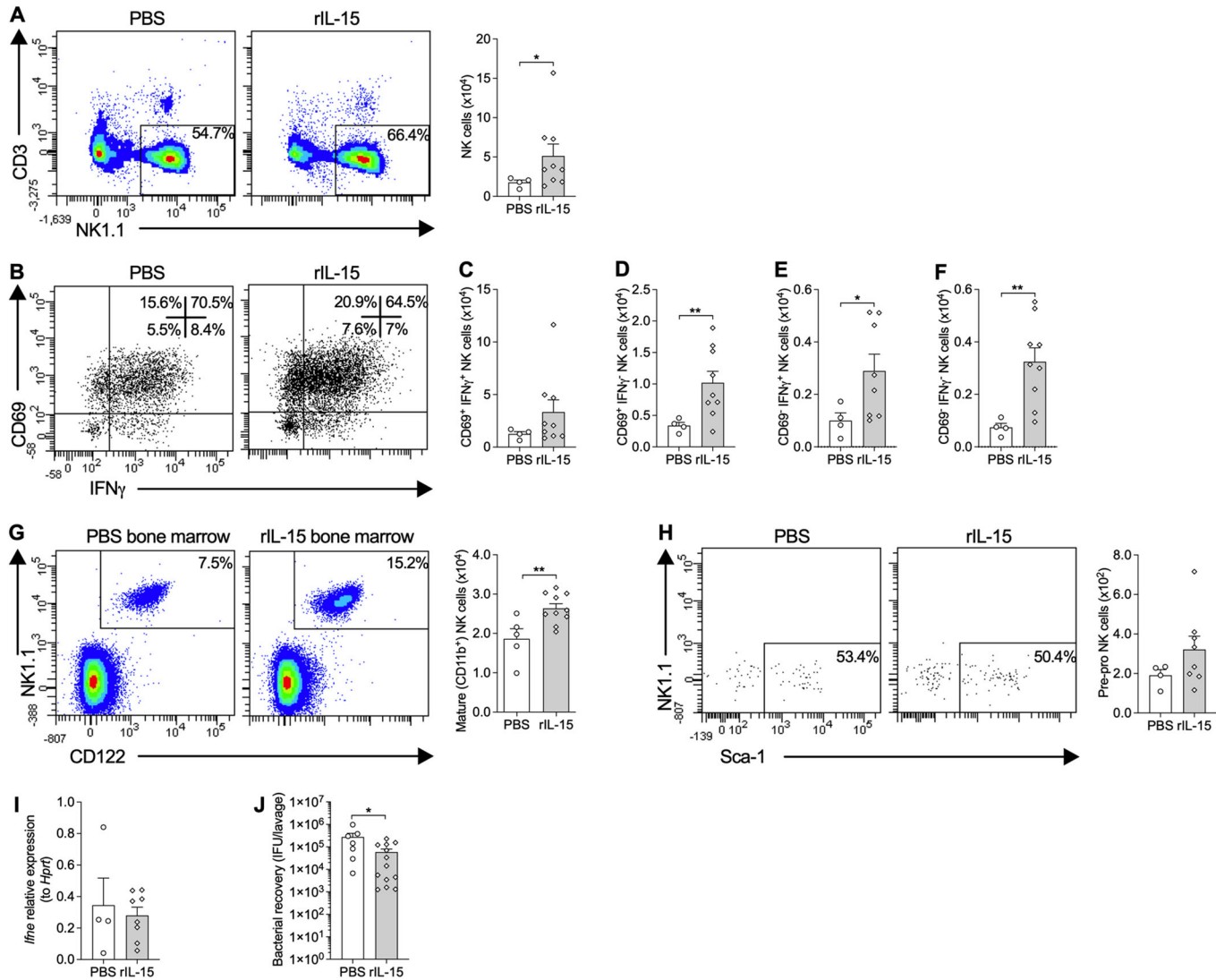

**Figure 6. Local administration of recombinant (r)IL-15 increases NK cells in the female uterus and bone marrow and decreases _Chlamydia_ infection, but does not alter pre-pro NK cell progenitors or interferon (IFN)ε expression.**

(A) Flow cytometry of uterine horn cells from wild-type (WT) C57BL/6 mice prophylactically administered rIL-15 or 0.1% bovine serum albumin (BSA) in phosphate-buffered saline (PBS) vehicle control transcervically, on day 3 of _Chlamydia muridarum_ infection showing conventional NK cells (FSC$^{low-int}$ SSC$^{low}$ CD45$^+$ CD3$^-$ NK1.1$^+$) and quantification. (B) Flow cytometry of uterine horn cells as in (A), showing CD69$^{+/-}$ IFNγ$^{+/-}$ conventional NK cells. (C–F) Quantification of (C) CD69$^+$ IFNγ$^+$, (D) CD69$^+$ IFNγ$^-$, (E) CD69$^-$ IFNγ$^+$, and (F) CD69$^-$ IFNγ$^-$ conventional NK cells in uterine horns as in (B). (G) Flow cytometry of bone marrow from femurs as in (A), showing mature NK cells (FSC$^{low-int}$ SSC$^{low}$ CD45$^+$ lin$^-$ CD11b$^+$ CD122$^+$ NK1.1$^+$) and quantification. (H) Flow cytometry of uterine horn cells as in (A), showing pre-pro NK cell progenitors (FSC$^{low-int}$ SSC$^{low}$ CD45$^+$ lin$^-$ FLT3$^-$ IL-7Rα$^+$ C-kit$^{low/-}$ CD122$^-$ NK1.1$^-$ CD49b$^-$ NKG2D$^+$ Sca-1$^+$) and quantification. (I) qPCR analysis of _Ifne_ expression, normalized to the expression of the housekeeping gene _Hprt_, in uterine horns as in (A). (J) qPCR analysis of _Chlamydia_ numbers normalized to inclusion forming units (ifu) standards in vaginal lavage fluid as in (A). Data information: The % displayed on the flow cytometry plots are the % of the parent population the cells within the gates/quadrants comprise. All data presented as mean ± SEM, with individual values. *$p < 0.05$, **$p < 0.01$ ((A, C–J): one-tailed Mann–Whitney test). ((A–F, H–I): $n \geq 4$, (G): $n \geq 5$, (J): $n \geq 8$ (all data from one experiment; biological replicates). For flow cytometry on uterine tissue, uteri from 1 to 3 mice were pooled for each biological replicate and at least 4 pooled samples were analyzed. See also Fig. EV4. Source data are available online for this figure.

expression profiles of these cells in the non-pregnant mouse uterus are few. One limitation of our current study is that we do not identify whether IFNε-mediated increases in IL-15 production are due to increases in the recruitment and/or proliferation of IL-15 producing cells in the uterus or increases in the amount of IL-15 produced per cell. Further investigation is required to determine the role of IFNε in mediating IL-15 responses by monocytes/macrophages in the non-pregnant uterus.

Consistent with our previous studies showing IFNε regulation of CD69 expression on NK cells in whole splenocytes stimulated ex vivo (Stifter et al, 2018), we show that IFNε alone directly stimulates NK cell activation (CD69 expression). However, IL-15 is required to increase NK cell number, IFNγ production, and cytolytic activity. Increases in NK cell number observed with 18 h IL-15 stimulation are likely due to increased survival, rather than proliferation, as others showed that the time to first division in

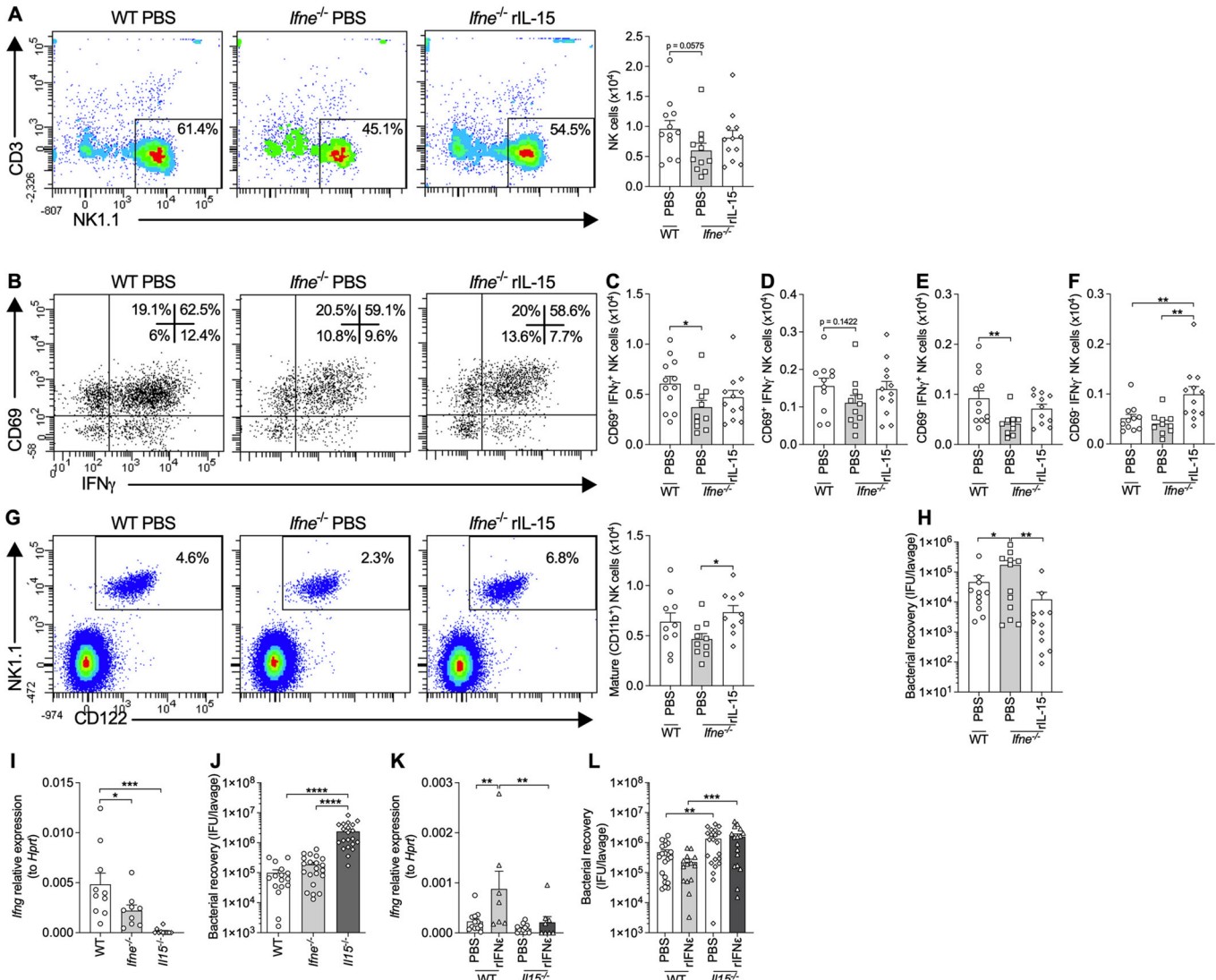

**Figure 7. Local administration of recombinant (r)IL-15 to *Ifne*-/- mice partially restores NK cell populations in the uterus and bone marrow and decreases *Chlamydia* infection, while rIFNε administration in *Il15*-/- mice does not restore protective responses.**

(A) Flow cytometry of uterine horn cells from *Ifne*-/- and wild-type (WT) C57BL/6 mice prophylactically administered rIL-15 (*Ifne*-/-) or 0.1% bovine serum albumin (BSA) in phosphate-buffered saline (PBS) vehicle control (WT and *Ifne*-/-) transcervically, on day 3 of *Chlamydia muridarum* infection showing conventional NK cells (FSC^low-int SSC^low CD45+ CD3- NK1.1+) and quantification. (B) Flow cytometry of uterine horn cells as in (A), showing CD69+/- IFNγ+/- conventional NK cells. (C–F) Quantification of (C) CD69+ IFNγ+, (D) CD69+ IFNγ-, (E) CD69- IFNγ+, and (F) CD69- IFNγ- conventional NK cells in uterine horns as in (B). (G) Flow cytometry of bone marrow from femurs as in (A), showing mature NK cells (FSC^low-int SSC^low CD45+ lin- CD11b+ CD122+ NK1.1+) and quantification. (H) qPCR analysis of *Chlamydia* numbers normalized to inclusion forming unit (ifu) standards in vaginal lavage fluid from mice as in (A). (I) qPCR analysis of *Ifng* expression, normalized to the expression of the housekeeping gene *Hprt*, in uterine horns from *Il15*-/-, *Ifne*-/-, and WT C57BL/6 mice on day 3 of *C. muridarum* infection. (J) qPCR analysis of *Chlamydia* numbers normalized to ifu standards in vaginal lavage fluid from mice as in (I). (K) qPCR analysis of *Ifng* expression in uterine horns from *Il15*-/- and WT C57BL/6 mice prophylactically administered rIFNε or PBS vehicle control transcervically, on day 3 of *C. muridarum* infection. (L) qPCR analysis of *Chlamydia* numbers normalized to ifu standards in vaginal lavage fluid from mice as in (K). Data information: The % displayed on the flow cytometry plots are the % of the parent population the cells within the gates/quadrants comprise. All data presented as mean ± SEM, with individual values. *$p < 0.05$, **$p < 0.01$, ***$p < 0.001$, ****$p < 0.0001$ ((A, C–L): one-way ANOVA). (A–H): $n \geq 10$ (data from one experiment), (I–J): $n \geq 9$ (data from one experiment), (K): $n \geq 7$ (data from one experiment), (L): $n \geq 14$ (data from two experiments; all biological replicates). For flow cytometry on uterine tissue, uteri from 1 to 3 mice were pooled for each biological replicate and at least 4 pooled samples were analyzed. See also Fig. EV5. Source data are available online for this figure.

response to IL-15 is ~32.5 h (Zhao and French, 2012). Our results suggest that IFNε does not directly affect NK cell survival but contributes to NK cell accumulation *via* the actions of IL-15. CD69 is both an early marker of NK cell activation induced by various stimuli including type I IFNs and bacteria (Athié-Morales et al, 2008) and an activating receptor that, when ligated, induces NK cell

cytolytic activity, proliferation, adhesion molecule expression, and cytokine production that sustain NK cell responses (Borrego et al, 1999). Thus, the increase in CD69 surface expression by IFNε may also contribute to these NK cell functions in the presence of CD69 ligands, such as galectin-1 on the surface of macrophages and DCs in vivo, however, in monoculture, IFNε stimulation alone is unable

to increase IFNγ production or cytolytic activity. Notably, co-stimulation with IFNε and IL-15 had synergistic effects, indicating that they may act in concert to maximize NK cell responses. The effects of rIFNε stimulation in combination with anti-IL-15 neutralizing antibody were like those of rIFNε alone, demonstrating that the direct effects of IFNε on NK cell activation are independent of autocrine IL-15 production, and confirming that NK cells themselves are not the source of IFNε-induced IL-15. (Souza-Fonseca-Guimaraes et al, 2012).

We show that rIFNε and rIL-15 have similar effects on NK cell accumulation during infection in WT mice, consistent with IFNε mediating this NK cell response through the actions of IL-15 in vivo. Interestingly, we also show that rIFNε promotes the accumulation of pre-pro NK cell progenitors independent of IL-15. Pre-pro NK cells are characterized by the absence of CD122, a component of the IL-15 receptor, and so are unable to respond to IL-15 at this differentiation stage (Fathman et al, 2011; Goh and Huntington, 2017). Local administration of rIL-15 increased the production of mature NK cells in the bone marrow during infection but rIFNε failed to replicate this. This may be due to the requirement for IL-15 production downstream of IFNε for the production of mature NK cells and insufficient time between rIFNε treatment and analysis to allow these processes to occur systemically. rIL-15 treatment in vivo had a modest effect in reducing Chlamydia burden compared to the effect of complete genetic ablation of IL-15. This is likely due to the degree of loss/gain of function in these experiments and rIL-15 not reaching the target cells and/or for a sufficient duration. Higher doses of rIL-15 or alternate routes of administration may be required to induce stronger protective NK cell IFNγ responses in vivo and have a greater impact on Chlamydia burden. In Ifne[-/-] mice, local administration of rIL-15 prior to infection increases inactive NK cell numbers in the uterus and the production of mature NK cells in the bone marrow, but has no effect on active and IFNγ-producing populations locally. This indicates that, in vivo, the presence of IFNε and its direct IL-15-independent effects are required to potently induce NK cell activation locally.

Our findings demonstrating direct, IL-15-independent effects of IFNε on NK cells are consistent with reports that NK cell expression of the type I IFN receptor (IFNAR) is required to mediate their cytolytic activity and IFNγ production in other sites during viral infections (Gill et al, 2011; Martinez et al, 2008; Zhu et al, 2008). However, these studies do not identify which of the 17 type I IFNs act directly on NK cells for these responses. Similarly, our data showing IL-15-dependent effects of IFNε on NK cells are supported by reports that, downstream of IFNAR signaling, specific NK cell functions are mediated through the actions of IL-15 (Lucas et al, 2007) but these studies also do not identify the type I IFN responsible.

This study used C. muridarum in mice to define the relation-ships between IFNε and NK cell responses in the uterus. The advantage of using C. muridarum in murine models is that it is the natural mouse pathogen, making it the most appropriate serovar for investigating host-pathogen relationships in the FRT. This is especially true for the study of acute infection, protective immunity, and pathogenesis (De Clercq et al, 2013; Morrison and Caldwell, 2002). In mice a low inoculum of C. muridarum establishes a productive infection that ascends the FRT and induces sustained endometrial inflammation and tubal pathology in immune-

sufficient mice, and therefore, it reliably recapitulates the course of C. trachomatis FRT infections observed in women. C. trachomatis is not a natural pathogen in mice, and in immune-sufficient animals, it does not ascend the FRT beyond the cervix. This is unless the cervix is bypassed by infecting via the intrauterine or intrabursal route (Gondek et al, 2012; Tuffrey et al, 1990) or extremely large inocula are administered (~$10^7$ IFU) (Darville et al, 1997). This bacterium also does not induce the cardinal features of Chlamydia-induced disease in women (Darville et al, 1997; Gondek et al, 2012). Furthermore, IFNε is expressed by epithelial cells throughout the FRT but most strongly in the uterine epithelium, therefore, for these studies we used C. muridarum that reliably ascends into the epithelium of the upper reproductive tract to investigate the regulation of immune responses by IFNε.

IFNγ is associated with protection in both C. muridarum and C. trachomatis infections, however, Chlamydia species do differ in their susceptibility to IFNγ-mediated immune responses, with C. muridarum being more resistant to IFNγ-mediated effects (Morrison, 2000; Perry et al, 1999). This is a limitation of our study, as IFNε-mediated NK cell IFNγ responses may have a greater impact on C. trachomatis infection and suggests that C. muridarum and C. trachomatis could also differ in their resistance to other cytokines. Future studies are needed to determine if the novel IFNε-mediated immune responses we describe during C. muridarum infection have a similar impact on C. trachomatis, and other Chlamydia spp. and FRT pathogens. We also have not determined the relative contribution of NK cell responses in long-term sequelae. Such studies would require substantial optimization and additional animal experimentation. These are beyond the scope of the current study that shows a novel functional relationship between IFNε, IL-15 and NK cells in FRT host defense for the first time.

It would also be informative to assess IL-15 sufficient but NK cell deficient mice to further assess any NK cell independent effects of IFNε via other resident tissue or early innate immune cells. However, collectively our data show that NK cells are the main contributors to IFNε-mediated protective responses. We have previously shown the direct effects of IFNε on epithelial cells (Stifter et al, 2018), and here we show that numbers of other common immune cell populations in the uterus are unaffected by IFNε deficiency, and use Il15[-/-] mice, that are devoid of mature NK cells, to determine the NK cell-independent effects of IFNε in vivo. IL-15 deficiency may have effects on other cell types (Il15[-/-] mice also exhibit defects in NKT, intraepithelial lymphocytes and memory CD8 T cells) and so these mice are not exclusively NK cell deficient (Huntington, 2014). However, all currently available tools for depleting NK cells have non-specific effects and there are no markers expressed exclusively by NK cells to target with antibody depletion that are not also expressed by other cell types. Genetic deletion of factors required for NK cell development and function commonly affect T and B cell function, leads to the development of severe autoimmune disorders, or only target cytolytic activity (Zamora et al, 2015). Thus, Il15[-/-] mice are the most appropriate tool available to determine if there are NK cell-independent effects of IFNε on infection and protective responses.

Endometrial cells in the upper FRT are histologically similar between humans and mice, consisting of simple columnar luminal epithelium and tubular glands within a specialized stroma (Rendi et al, 2012; Yamaguchi et al, 2021). To add further clinical relevance to our findings we performed correlation analysis on endometrial

biopsies. We show that in women, IFNε expression is linked to both the expression of IL-15 and NCR1, a marker of NK cell recruitment. It was not possible to obtain endometrial tissues at early time points during infection due to ethical and logistical reasons, and studies in human tissues ex vivo do not incorporate the influx of immune cells from the circulation or the effects on bone marrow haematopoiesis, which are key responses to infection.

In conclusion, regulation of NK cell responses in the uterus is crucial for maintaining FRT function and protection against infection. As a result, the regulation of NK cells in the uterus is unique. To date, no studies have identified uterine-specific factors that control NK cell responses in this immune-privileged site in response to infection. To the best of our knowledge this manuscript is the first to identify a crucial role for the constitutive expression of a uterine-specific factor, IFNε, in controlling NK cell responses in the uterus and protecting against FRT infections. Since the uterus is an immunologically privileged organ that is dependent on NK cell responses for its reproductive functions, the ramifications of our findings may reach beyond protection against infection.

# Methods

## Mice

WT and *Ifne*[-/-] C57BL/6 mice were purchased from Australian BioResources or Monash University, Australia. *Il15*[-/-] (and matched WT controls) and IL-15-CFP mice were from the Walter and Eliza Hall Institute of Medical Research (WEHI), Australia. Mice were housed 4 per cage in individually ventilated and filtered cages under positive pressure in an SPF facility on a 12-h light:dark cycle with controlled temperature ($22 \pm 2$ °C) and humidity (30-70%). Each cage was equipped with autoclaved corn cob bedding, a shelter, nesting paper, paper coils, and a wooden tongue depressor for environmental enrichment. Mice acclimatized for at least 5 days prior to experimentation. All mice were fed a standard mouse diet available ad libitum (Specialty Feeds, WA, Australia). All animal procedures used in this study were performed in accordance with the recommendations set out in the Australian code of practice for the care and use of animals for scientific purposes issued by the National Health and Medical Research Council (Australia). All protocols were approved by the University of Newcastle (approval number: A-2011-109) and WEHI (approval number: 2018.002) Animal Care and Ethics Committees. Mice were monitored daily for signs of disease as per the approved protocol. Their weight, appearance, and behavior were within normal parameters before being assigned to groups. Intervention by veterinary treatment or euthanasia was indicated by the development of signs of severe disease. There were no interventions required as a result of our experimental protocol.

## *Chlamydia muridarum* infection

Mice were allocated into groups using a minimization strategy to balance variables, namely age and weight, across groups. Age matched, adult (6–8 weeks old), female WT, *Ifne*[-/-], or *Il15*[-/-] C57BL/6 mice were pre-treated with 2.5 mg depot medroxyprogesterone acetate (Depo-Provera; Pfizer, NY, USA) subcutaneously to prime for infection and synchronize their estrogenic cycles. Seven days later, mice were infected by intravaginal inoculation with $5 \times 10^4$

IFU *C. muridarum* (ATCC VR-123) in 10 μL sucrose-phosphate-glutamate buffer (SPG; 10 mM sodium phosphate [pH 7.2], 0.25 M sucrose, 5mM L-glutamic acid) or sham-infected with SPG alone under ketamine:xylazine anesthesia (80 mg/kg:5 mg/kg IP; Ilium Ketamil® and Ilium Xylazil-20®; Troy Laboratories, Glendenning, Australia), as described previously (Asquith et al, 2011; Fung et al, 2013; Fig. EV4A). Mice were sacrificed by sodium pentobarbital (250 mg/kg IP; Lethabarb; Virbac, Milperra, NSW, Australia) overdose at 3dpi for characterization of immune responses or 14dpi for quantification of FRT pathology.

## In vivo administration of recombinant cytokines

Some groups of mice were administered recombinant mouse (r) IFNε ($1.36 \times 10^5$ IU/mg; generated in-house as described previously (Fung et al, 2013)), rIL-15 (R&D Systems, Minneapolis, MN, USA), or vehicle transcervically 24 h before and again intravaginally 6 h before infection. For transcervical treatments, mice were anesthetized with isoflurane (1.5–5% in $O_2$) and 100 μL rIFNε (7 μg) in PBS, rIL-15 (285 ng) in PBS with 1% BSA, or either vehicle alone was injected into the uterine lumen using a small animal endoscope with a working sheath and flexible needle (Mainz COLOVIEW® System, Karl Storz, Tuttlingen, Germany). For intravaginal treatments, mice were anesthetized with isoflurane and 20 μL rIFNε (3 μg) in PBS, rIL-15 (285 ng) in 1% BSA in PBS, or either vehicle alone was deposited into the vaginal vault using a pipette (Fig. EV4A,B). Specialist reagents are available subject to stocks.

## Flow cytometry

Uterine horn tissue was processed into single-cell suspensions by gently dissociating in HEPES buffer (10 mM HEPES-NaOH [pH7.4], 150 mM NaCl, 5 mM KCI, 1 mM $MgCl_2$, 1.8 mM $CaCl_2$) and enzymatic digestion with collagenase-D (2 mg/mL; Roche) and DNase I (40 U/mL; Roche; 37 °C, 30 min). Spleens and lumbar aortic and medial iliac lymph nodes were dissociated by passing through a sterile sieve. Bone marrow was obtained by flushing the left femur with 1 mL FACS buffer (2% fetal bovine serum [FBS], 2 mM EDTA in PBS). Debris was removed from single cell suspensions using a 70 μm nylon cell strainer. Erythrocytes were removed by incubation with red blood cell lysis buffer (155 mM $NH_4Cl$, 12 mM $NaHCO_3$, 0.1 mM ethylenediaminetetraacetic acid [EDTA], pH 7.35; 5 min, 4 °C) and cells enumerated using a Countess™ automated cell counter (Invitrogen).

Cells ($0.5–3 \times 10^6$ cells/sample) were incubated with mouse Fc receptor block (anti-mouse CD16/32; BioXCell; 10 ng/mL, 15 min, 4 °C). Cell surface markers (CD45, CD3, NK1.1, CD122, CD11b, CD49b, NKG2D, CD4, B220, GR1, C-kit, Sca-1, FLT3, IL-7Rα, and/or CD69) were labeled by incubation with fluorochrome or biotin-conjugated antibodies (Table EV1; 20 min, 4 °C). Biotin-conjugated antibodies were subsequently labeled with streptavidin-conjugated fluorochromes. Stained cells were fixed in 4% paraformaldehyde in PBS and analyzed using a FACSCanto™ II or FACSAria™ III flow cytometer and FACSDiva software (BD Biosciences) (Beckett et al, 2012). After exclusion of debris, cell populations were identified based on forward- and side-scatter and characteristic surface marker expression profile (Table EV2; gating strategy shown in Fig. EV1) and proportion (as a percentage) and total numbers in each tissue calculated (Beckett et al, 2012; Yadi et al, 2008).

For cytokine detection, single cell suspensions were incubated with ionomycin (1 µg/mL), phorbol 12-myristate 13-acetate (PMA; 50 ng/mL), and Brefeldin A (5 µg/mL; Sigma-Aldrich), in supplemented RPMI 1640 media (10% FBS, 2 mM L-glutamine, 2 mM sodium pyruvate, 100 µg/mL penicillin, 100 µg/mL streptomycin, 50 µM 2-mercaptoethanol; Gibco®, Thermo Fisher Scientific; 5 h, 37 °C) prior to staining for surface markers (Essilfie et al, 2012). Cells were then fixed in 4% paraformaldehyde (30 min; 4 °C), permeabilized with saponin buffer (0.25% saponin in FACS buffer; 10 min), and stained with PE- or BV711-conjugated anti-IFNγ (Table EV1; 30 min). Control samples were stained with isotype-matched control antibody to assist in analysis.

To detect IL-15 and IFNε expressing cells, uterine horn tissue from IL-15-CFP reporter mice was incubated in HBSS (Gibco) supplemented with 2% FBS (Bovogen Biologicals) and 2.5 mM EDTA (30 min) followed by digestion in RPMI supplemented with 5% FBS, 1 mg/mL Collagenase type III (Scimar) and 2 µg/mL DNase I (Sigma; 90 min). Single-cell suspensions were prepared by passing through a 70 µm filter (Falcon). Uterine single cells were incubated with Brefeldin A (1 µg/mL; Sigma-Aldrich) in supplemented RPMI 1640 media (4 h, 37 °C) prior to staining. Viability staining was performed with Zombie Red fixable viability dye (Biolegend), prior to performing Fc block and surface stain using BV510-labeled anti-CD45, APC Cy7-labeled anti-CD11b, PE Cy7-labeled anti-CD11c, Pacific Blue-labeled anti-Ly6G, AF700-labeled anti-MHC-II, BV650-labeled anti-CD19, and BV711-labeled anti-NKp46, or BV785-labeled anti-CD4 and APC-Cy7-labeled anti-CD8 (Table EV1). Cells were fixed and permeabilized with the Foxp3/Transcription Factor Staining Buffer Set (eBioscience), and stained with APC-labeled anti-IFNε monoclonal IgG2a (generated in-house (Fung et al, 2013)), PE-labeled anti-pan Cytokeratin and AF488-labeled anti-GFP. Fluorescence minus one controls were used to assist in analysis. Samples were analyzed using a Fortessa X20 flow cytometer and FlowJo V10 software (BD).

## Gross oviduct pathology

In mice, the extent of swelling of the oviducts has been shown to be proportional to the severity of disease (O'Meara et al, 2014; Shah et al, 2005). FRT pathology was quantified by measuring the longitudinal and transverse diameters of the oviduct using digital calipers and multiplying these values to determine the oviduct cross-sectional area in mm$^2$.

## Immunofluorescence

Uteri from IL-15-CFP reporter mice were fixed in 10% neutral buffered formalin and embedded in paraffin. Four µm sections were cut and deparaffinized routinely. Heat-induced epitope retrieval (HIER) was performed in citrate buffer (10 mM trisodium citrate, pH 6.0) under pressure using a Biocare decloaking chamber (Biocare; 5 min, 110 °C). To detect IFNε protein, sections were labeled with mouse anti-IFNε (3 µg/mL; generated in-house (Fung et al, 2013)), biotinylated anti-mouse IgG secondary antibody (Table EV1), followed by AF594-labeled Streptavidin (Invitrogen). Background staining was prevented using goat serum (Vector Laboratories). CFP expression was detected with AF488-labeled anti-GFP. Slides were counterstained with 4',6-diamidino-2'-phenylindole dihydrochloride (DAPI; Molecular Probes) and

coverslipped with Fluorescence mounting medium (Agilent). Slides were scanned at 20x magnification using a VS120 Virtual Slide Microscope (Olympus) and analyzed with OlyVIA software (Olympus).

## ELISA

Concentrations of IFNγ and IL-15 were measured in uterine lavage fluid (flushed with 100 µL PBS supplemented with cOmplete™ Mini protease and phosSTOP phosphatase inhibitor cocktails, Sigma-Aldrich) by ELISA (R&D Systems), as per the manufacturer's instructions.

## Human endometrial tissue collection

Human protocols were approved by the University of Newcastle Human Research Ethics Committee (approval number: H-2022-0202) and conformed to the principles set out in the WMA Declaration of Helsinki and the Department of Health and Human Services Belmont Report. Informed consent was obtained from all human subjects as per approved guidelines. Normal endometrial tissue was collected from patients (mean age, 66.23; SD, 11.46) undergoing surgical interventions for uterine cancer at the John Hunter Hospital, Newcastle. A pathologist confirmed the suitability of collected samples by histopathology. Tissues were collected in HBSS, transported on ice, and extensively washed in PBS to remove excessive blood then stored in LN$_2$ until use.

## Gene expression analysis

Total RNA was extracted from homogenized uterine tissue using TRIzol® Reagent (Thermo Fisher Scientific) according to the manufacturer's instructions. Concentration and purity of RNA was quantified using a NanoDrop™ 1000 Spectrophotometer (Thermo Fisher Scientific). RNA (1 µg) was treated with DNase I (Sigma-Aldrich) and reverse transcribed using BioScript™ reverse transcriptase enzyme and random hexamer primers (Bioline), according to the manufacturer's instructions (Starkey et al, 2013). Gene expression was evaluated by real-time qPCR using custom designed primers (Table EV3; IDT) and SYBR Green Supermix (KAPA Biosystems), or TaqMan® Gene Expression Assays (mIfne: Mm00616542_s1; mHprt: Mm00446968_m1; hIfne: Hs00703565_s1; hHprt1: Hs02800695_m1; Thermo Fisher Scientific), on a Mastercycler® ep realplex2 system (Eppendorf) (Horvat et al, 2010; Phipps et al, 2007). Expression levels of genes were calculated relative to the reference gene using the $2^{-\Delta\Delta Ct}$ method. Crucial qPCR data (e.g., Il15 expression in uterine tissue) was confirmed by assessing protein levels in uterine lavage fluid by ELISA (Fig. 3B).

## NK cell isolation and stimulation

Splenic NK cells were isolated from single-cell suspensions of splenocytes from WT female C57BL/6 mice using the EasySep™ Mouse CD49b Positive Selection Kit, which simultaneously labels the cNK cell marker, CD49b, and EasyEights™ EasySep™ Magnet (Stemcell Technologies, Vancouver, BC, Canada). Isolated NK cells were then cultured in supplemented RPMI 1640 media containing 100 pM/mL rIFNε (Stifter et al, 2018), 100 ng/mL rIL-15, 10 µg/mL anti-IL-15 (#ab7213; Abcam, UK) and/or isotype control antibody (#ab171870;

**The paper explained**

**Problem**

The female reproductive tract (FRT) is a unique mucosal site where the regulation of immune responses is finely balanced to be permissive of a semi-autologous fetus yet protective against sexually transmitted infections (STIs) such as *Chlamydia*, HSV and HIV. The regulation of NK cells, in health and in response to infection, in this location is unique. To date, no studies have identified uterine-specific factors that control NK cell responses in this immune-privileged site in response to infection.

**Results**

Here, we demonstrate that the constitutively expressed type I IFN, IFNε, is a critical regulator of NK cell responses in the uterus. We show that IFNε promotes NK cell accumulation, activation, and effector cytokine production in response to *Chlamydia* infection through IL-15-dependent and -independent mechanisms. Even small changes in these responses would have large ramifications in the often chronic infections of the FRT.

**Impact**

This study is the first to identify a crucial role for the constitutive expression of a uterine-specific factor, IFNε, in controlling NK cell responses in the uterus and protecting against FRT infections. Since the uterus is an immunologically privileged organ that is dependent on NK cell responses for its reproductive functions, the ramifications of our findings may reach beyond protection against *Chlamydia* infection. These mechanisms are likely to have broader significance in understanding and manipulating FRT immunity to other STIs (e.g., HIV, HSV, HPV) and gynecological cancers, where similar regulation of protective immunity is required, and in NK cell-mediated maintenance of uterine function and physiology.

Abcam; Table EV1) for 18 h (37 °C, 5% $CO_2$). For intracellular cytokine detection, PMA, ionomycin, and Brefeldin A were added in the final 5 h of culture and cells analyzed by flow cytometry.

### Cytotoxicity assay

After stimulation, NK cells were co-cultured with YAC-1 target cells (mouse lymphoma lymphoblast fibroblast; ATCC TIB-160; authenticated and mycoplasma tested) at the indicated effector cell:target cell (E:T) ratios for 4 h (37 °C and 5% $CO_2$). Co-cultures were then stained with Fixable Viability Dye™ (FVD) eFluor™ 506 (eBioscience) and fluorochrome-conjugated antibodies specific for cell surface markers (CD45, CD3 and NK1.1) to differentiate NK and YAC-1 cells and analyzed by flow cytometry. NK cell only and YAC-1 only controls were used to determine spontaneous lysis of target cells and set gating strategy. Percent specific lysis was calculated as the proportion of YAC-1 cells ($FSC^{int}$ $SSC^{int-hi}$ $CD45^+$ $CD3^+$ $CD49b^-$) positive for FVD, minus spontaneous lysis (Littwitz-Salomon et al, 2018; Valiathan et al, 2012).

### *Chlamydia* load

Vaginal lavage fluid was collected from mice by lavaging the vagina with sterile SPG (2 ×60 μL). Total DNA was extracted using a GF-1 Bacterial DNA Extraction Kit (Vivantis, Malaysia), according to the manufacturer's instructions. *Chlamydia* numbers (in IFU) were then determined by SYBR Green-based real-time qPCR using

primers specific for genomic *C. muridarum ompA* (Table EV3) and standards of known concentrations of *C. muridarum* (determined by infection of McCoy cells), as described previously (Asquith et al, 2011; Berry et al, 2004; Fung et al, 2013; Horvat et al, 2010; Kaiko et al, 2008).

### Statistics

Investigators were blinded to experimental groups during analysis of samples. Sample size was chosen based on power calculations (80% power, alpha=0.05) using means and SDs of NK cell frequencies from pilot experiment shown in Fig. EV1A. Data are presented as mean ± SEM and/or individual values, as indicated. Outlier testing (ROUT) was performed on all datasets and any statistical outliers removed. Statistical significance for comparisons between two groups was determined using an unpaired Mann-Whitney test, two-tailed or one-tailed, where appropriate. Statistical significance for comparisons involving three or more groups was determined by one-way ANOVA with Fisher's Least Significant Difference (LSD) post hoc test. Statistical analyses were performed using GraphPad Prism 6 software (San Diego, CA).

## Data availability

This study includes no data deposited in external repositories.

## Peer review information

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

## Acknowledgements

This work was funded by grants from the National Health and Medical Research Council (NHMRC) of Australia to PMH, PJH, and JCH (1059242, 1003591) and by MJB via the Hunter New England Local Health District (HNELHD). PMH is funded by a Fellowship and Investigator grant from the NHMRC (1079187, 1175134). We thank Tegan Hunter (research assistant, University of Newcastle), Nicole Cole (Central Analytical Facilities, University of Newcastle), and the staff of the University of Newcastle BioResearch Facilities for their technical assistance and support.

## Author contributions

**Jemma R Mayall**: Formal analysis; Investigation; Methodology; Writing—original draft; Project administration; Writing—review and editing **Jay C Horvat**: Conceptualization; Formal analysis; Supervision; Funding acquisition; Investigation; Methodology; Writing—original draft; Project administration; Writing—review and editing. **Niamh E Mangan**: Formal analysis; Supervision; Investigation; Methodology; Project administration; Writing—review and editing. **Anne Chevalier**: Formal analysis; Investigation; Methodology. **Huw McCarthy**: Formal analysis; Investigation; Methodology. **Daniel Hampsey**: Investigation; Methodology. **Chantal Donovan**: Investigation; Methodology. **Alexandra C Brown**: Investigation; Methodology. **Antony Y Matthews**: Investigation; Methodology. **Nicole A de Weerd**: Investigation; Methodology. **Eveline D de Geus**: Formal analysis; Investigation; Methodology; Writing—review and editing. **Malcolm R Starkey**: Formal analysis. **Richard Y Kim**: Formal analysis. **Katie Daly**: Formal analysis. **Bridie J Goggins**: Formal analysis. **Simon Keely**: Formal analysis. **Steven Maltby**: Formal analysis. **Rennay Baldwin**: Formal analysis. **Paul S Foster**: Supervision. **Michael J Boyle**: Conceptualization; Funding acquisition; Investigation; Methodology; Writing—review and editing. **Pradeep S Tanwar**: Formal analysis; Investigation; Methodology; Writing—original draft. **Nicholas D Huntington**: Conceptualization; Supervision; Investigation; Methodology; Writing —original draft; Project administration. **Paul J Hertzog**: Conceptualization; Formal analysis; Supervision; Funding acquisition; Investigation; Methodology; Writing—original draft; Project administration; Writing—review and editing. **Philip M Hansbro**: Conceptualization; Resources; Formal analysis; Supervision; Investigation; Methodology; Writing—original draft; Project administration; Writing—review and editing.

## Disclosure and competing interests statement

The authors declare no competing interests.

# Expanded View Figures

**Figure EV1.  Effect of interferon (IFN)ε deficiency on the frequency of leukocyte populations in the uterus at 3 days and oviduct pathology at 14 days post *Chlamydia* infection and flow cytometry gating strategy for identification of natural killer (NK) cell and progenitor populations.**

(A) Frequency of natural killer (NK) cells (FSC$^{low-int}$ SSC$^{low}$ CD3$^-$ NK1.1$^+$), macrophages (FSC$^{int}$ SSC$^{int}$ F480$^+$), neutrophils (FSC$^{low-int}$ SSC$^{int-high}$ F480$^-$ CD11b$^+$ GR1$^+$), plasmacytoid dendritic cells (pDCs; FSC$^{low-int}$ SSC$^{low-int}$ CD11c$^+$ CD11b$^-$ GR1$^+$ PDCA$^+$), myeloid (m)DCs (FSC$^{low-int}$ SSC$^{low-int}$ CD11c$^+$ CD11b$^+$ GR1$^-$ PDCA$^-$), CD4$^+$ T cells (FSC$^{low-int}$ SSC$^{low}$ CD3$^+$ CD4$^+$), CD8$^+$ T cells (FSC$^{low-int}$ SSC$^{low}$ CD3$^+$ CD8$^+$), B cells (FSC$^{low-int}$ SSC$^{low}$ CD3$^-$ B220$^+$), and NK T cells (FSC$^{low-int}$ SSC$^{low}$ CD3$^+$ NK1.1$^+$) in uterine horns from *Ifne$^{-/-}$* and wild-type (WT) C57BL/6 mice on day 3 of *Chlamydia muridarum* infection measured by flow cytometry. (B) Flow cytometry of uterine horn cells from *Ifne$^{-/-}$* and WT mice on day 3 of *C. muridarum* infection showing gating for innate lymphoid cell (ILC) populations (ILC1-3: FSC$^{low-int}$ SSC$^{low}$ CD45$^+$ Lin$^-$ CD90.2$^+$ IL-7Rα$^+$ T-bet$^{+/-}$). (C) Cross-sectional area of the oviducts (in mm$^2$) from *Ifne$^{-/-}$* and WT mice on day 14 of *C. muridarum* (Cmu) or sham (SPG) infection. (D–H) Flow cytometry gating strategy for NK cell populations. (D) For all stains doublets and debris were first excluded, leukocytes (CD45$^+$ cells) selected and then lymphocytes selected based on size (forward scatter [FSC]) and granularity (side scatter [SSC]). (E) Conventional NK cells (FSC$^{low-int}$ SSC$^{low}$ CD45$^+$ CD3$^-$ NK1.1$^+$) and T cells (FSC$^{low-int}$ SSC$^{low}$ CD45$^+$ CD3$^+$ NK1.1$^-$) were gated based on NK1.1 and CD3 expression. (F) CD3$^-$ NK1.1$^-$ cells were gated and tissue-resident uterine (u)NK cells (FSC$^{low-int}$ SSC$^{low}$ CD45$^+$ CD3$^-$ NK1.1$^-$ CD49b$^-$ CD122$^+$) identified based on CD49b and CD122 expression. (G) Lineage marker (lin; CD3, CD4, B220, GR1, and CD11b)$^-$ cells were gated followed by FLT3$^-$ and IL-7Rα$^+$ C-kit$^{low/-}$ cells. CD122$^-$ NK1.1$^-$ cells were gated and pre-pro NK cell progenitors (FSC$^{low-int}$ SSC$^{low}$ CD45$^+$ lin$^-$ FLT3$^-$ IL-7Rα$^+$ C-kit$^{low/-}$ CD122$^-$ NK1.1$^-$ CD49b$^-$ NKG2D$^+$ Sca-1$^+$) gated based on CD49b, NKG2D and Sca-1 expression. CD122$^+$ NK1.1$^-$ cells were gated and precursor NK cell progenitors (FSC$^{low-int}$ SSC$^{low}$ CD45$^+$ lin$^-$ FLT3$^-$ IL-7Rα$^+$ C-kit$^{low/-}$ CD122$^+$ NK1.1$^-$ CD49b$^-$ NKG2D$^+$) gated based on CD49b and NKG2D expression. (H) In the bone marrow lin$^-$ CD11b$^-$cells were gated and immature NK cells (FSC$^{low-int}$ SSC$^{low}$ CD45$^+$ lin$^-$ CD11b$^-$ CD122$^+$ NK1.1$^+$) gated based on NK1.1 and CD122 expression. Lin$^-$ CD11b$^+$ cells were gated and mature NK cells (FSC$^{low-int}$ SSC$^{low}$ CD45$^+$ lin$^-$ CD11b$^+$ CD122$^+$ NK1.1$^+$) gated based on NK1.1 and CD122 expression. Data information: Data in (A) and (C) is presented as mean ± SEM, with individual values. $**p < 0.01$, $****p < 0.0001$ ((A): two-way ANOVA; (C): one-way ANOVA). (A): $n = 4$ (data from one experiment), (C): $n = $18-19 (data from two experiments; all biological replicates). For flow cytometry analysis of NK cells, single cell suspensions from (D–F) uterine horn, (D–F) spleen, (D, G) lymph node and (D, G, H) bone marrow tissues were stained with various antibodies (D, E) with or (D, F–H) without stimulation. (F) is repeated in Fig. 1B and pre-pro and precursor NK cell panels in (G) are repeated in Fig. 2A, E.

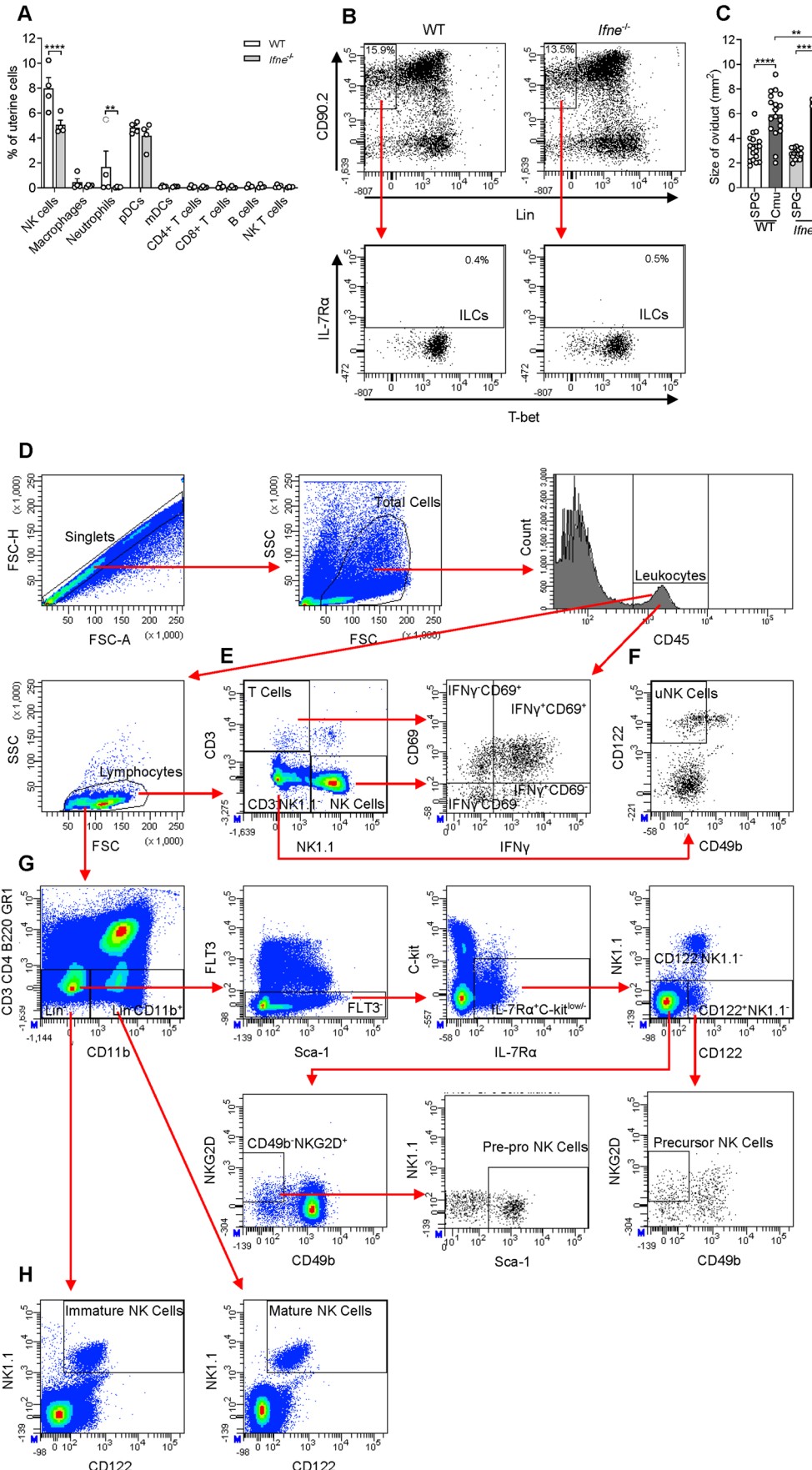

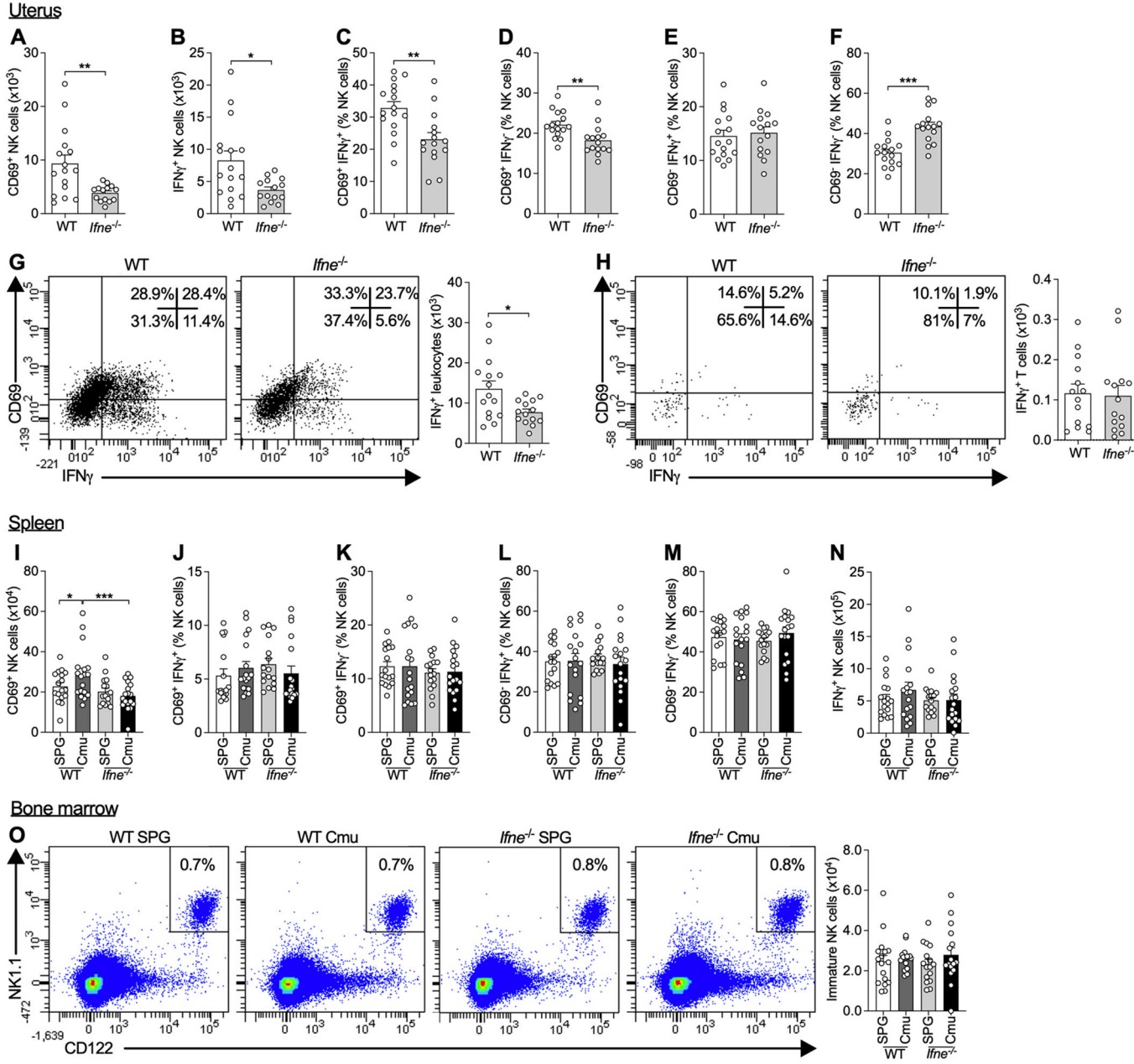

**Figure EV2. Effect of interferon (IFN)ε deficiency on the numbers and proportions of activated and IFNγ-producing natural killer (NK) cells in the uterus and spleen, numbers of IFNγ-producing T cells in the uterus, and numbers of immature NK cells in bone marrow.**

(A, B) Quantification of (A) CD69$^+$ and (B) IFNγ$^+$ conventional NK cells (FSC$^{low-int}$ SSC$^{low}$ CD45$^+$ CD3$^-$ NK1.1$^+$) in uterine horns from *Ifne$^{-/-}$* and wild-type (WT) C57BL/6 mice on day 3 of *Chlamydia muridarum* infection measured by flow cytometry. (C–F) Frequency of conventional NK cells expressing (C) CD69$^+$ IFNγ$^+$, (D) CD69$^+$ IFNγ$^-$, (E) CD69$^-$ IFNγ$^+$, and (F) CD69$^-$ IFNγ$^-$ in uterine horns. (G) Flow cytometry of uterine horn cells showing CD69$^{+/-}$ IFNγ$^{+/-}$ leukocytes (CD45$^+$) and quantification. (H) Flow cytometry of uterine horn cells showing CD69$^{+/-}$ IFNγ$^{+/-}$ T cells (FSC$^{low-int}$ SSC$^{low}$ CD45$^+$ CD3$^+$) and quantification. (I) Quantification of CD69$^+$ conventional NK cells (FSC$^{low-int}$ SSC$^{low}$ CD45$^+$ CD3$^-$ NK1.1$^+$) in spleens. (J–M) Frequency of conventional NK cells expressing (J) CD69$^+$ IFNγ$^+$, (K) CD69$^+$ IFNγ$^-$, (L) CD69$^-$ IFNγ$^+$, and (M) CD69$^-$ IFNγ$^-$ in spleens. (N) Quantification of IFNγ$^+$ conventional NK cells (FSC$^{low-int}$ SSC$^{low}$ CD45$^+$ CD3$^-$ NK1.1$^+$) in spleens. (O) Flow cytometry of bone marrow from femurs showing immature conventional NK cells (FSC$^{low-int}$ SSC$^{low}$ CD45$^+$ lin$^-$ CD11b$^-$ CD122$^+$ NK1.1$^+$) and quantification. Data information: The % displayed on the flow cytometry plots are the % of the parent population the cells within the gates/quadrants comprise. All data presented as mean ± SEM, with individual values. *$p < 0.05$, **$p < 0.01$, ***$p < 0.001$ ((A–H): two-tailed Mann–Whitney test; (I–O): one-way ANOVA). (A–H): $n \geq 15$ (data from three experiments), (I–O): $n \geq 15$ (data from two experiments; all biological replicates).

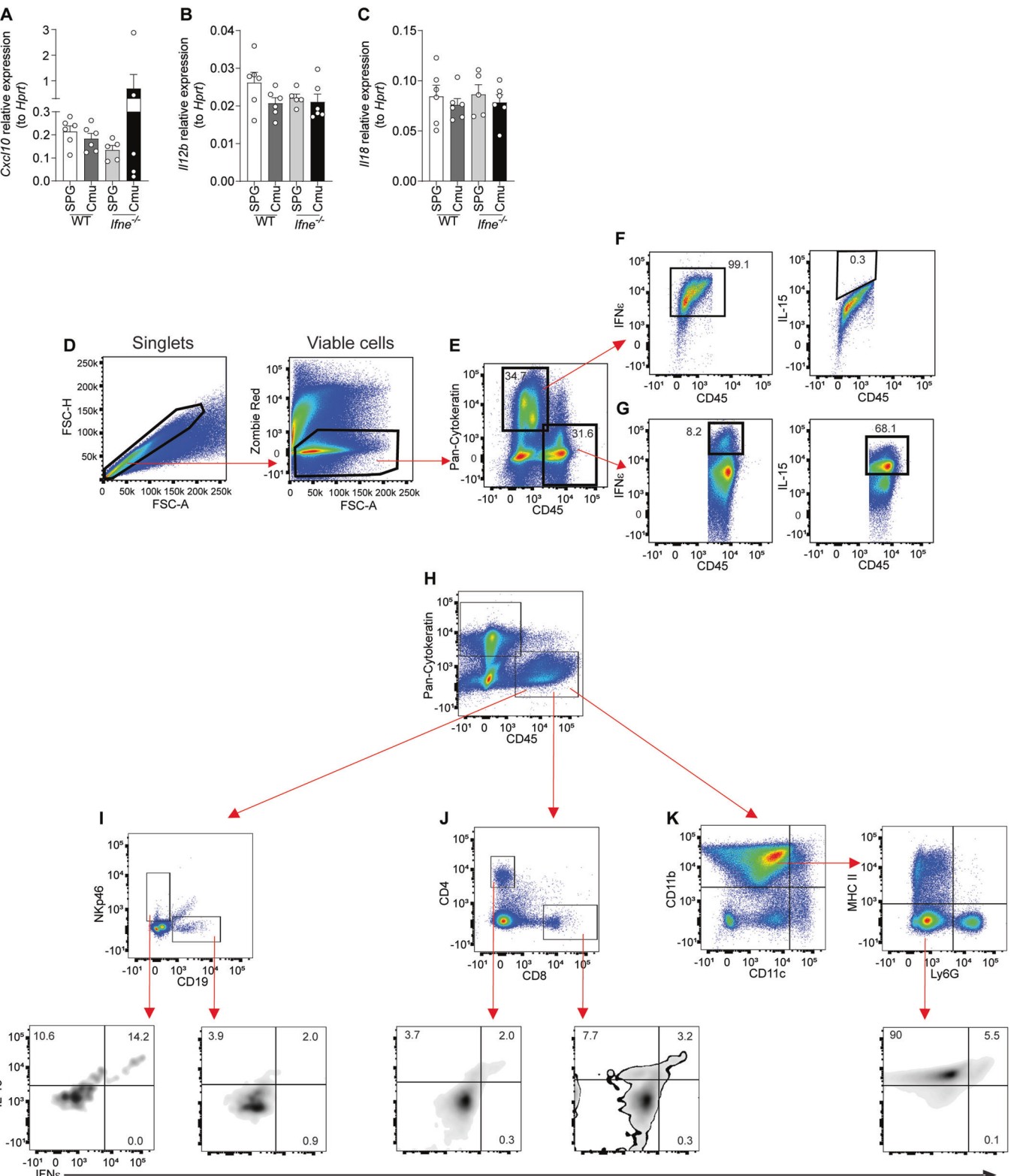

◀ **Figure EV3. Effect of interferon (IFN)ε deficiency on CXCL10, IL-12, and IL-18 expression in the uterus and flow cytometry gating strategy for identification of interferon (IFN)ε and IL-15 expressing cells in uteri and frequency of co-expression in common immune cell populations.**

(A–C) qPCR analysis of *Cxcl10* (A), *Il12b* (B) and *Il18* (C) expression normalized to the expression of the housekeeping gene *Hprt* in uterine horns from *Ifne*[-/-] and wild-type (WT) C57BL/6 mice on day 3 of *Chlamydia muridarum* (Cmu) or sham (SPG) infection. All data presented as mean ± SEM, with individual values (A-C: not significant, one-way ANOVA). (A–C): $n \geq 5$ (data from one experiment). (D) Single-cell suspensions of uteri from IL-15-CFP reporter mice were stained with antibodies against cell surface markers followed by intracellular staining. Single, viable cells were selected. (E) Epithelial cells (Pan-Cytokeratin[+]) and immune cells (CD45[+]) were gated based on Pan-Cytokeratin and CD45 expression. (F) IFNε and IL-15 expression was gated in epithelial cells and (G) immune cells, followed by identification of immune cell type shown in Fig. 3. (F, G) are repeated in Fig. 3G, H. (H–K) (H) To determine frequency of IFNε and IL-15 co-expression in immune cell subsets, CD45[+] cells were gated for (I) NK cells (NKp46[+]), B cells (CD19[+]), (J) T cells (CD4[+] or CD8[+]), and (K) monocytes/macrophages (CD11b[+] CD11c[neg-low] MHC-II[-] Ly6G[-]) then IFNε and IL-15 expression was characterized these populations.

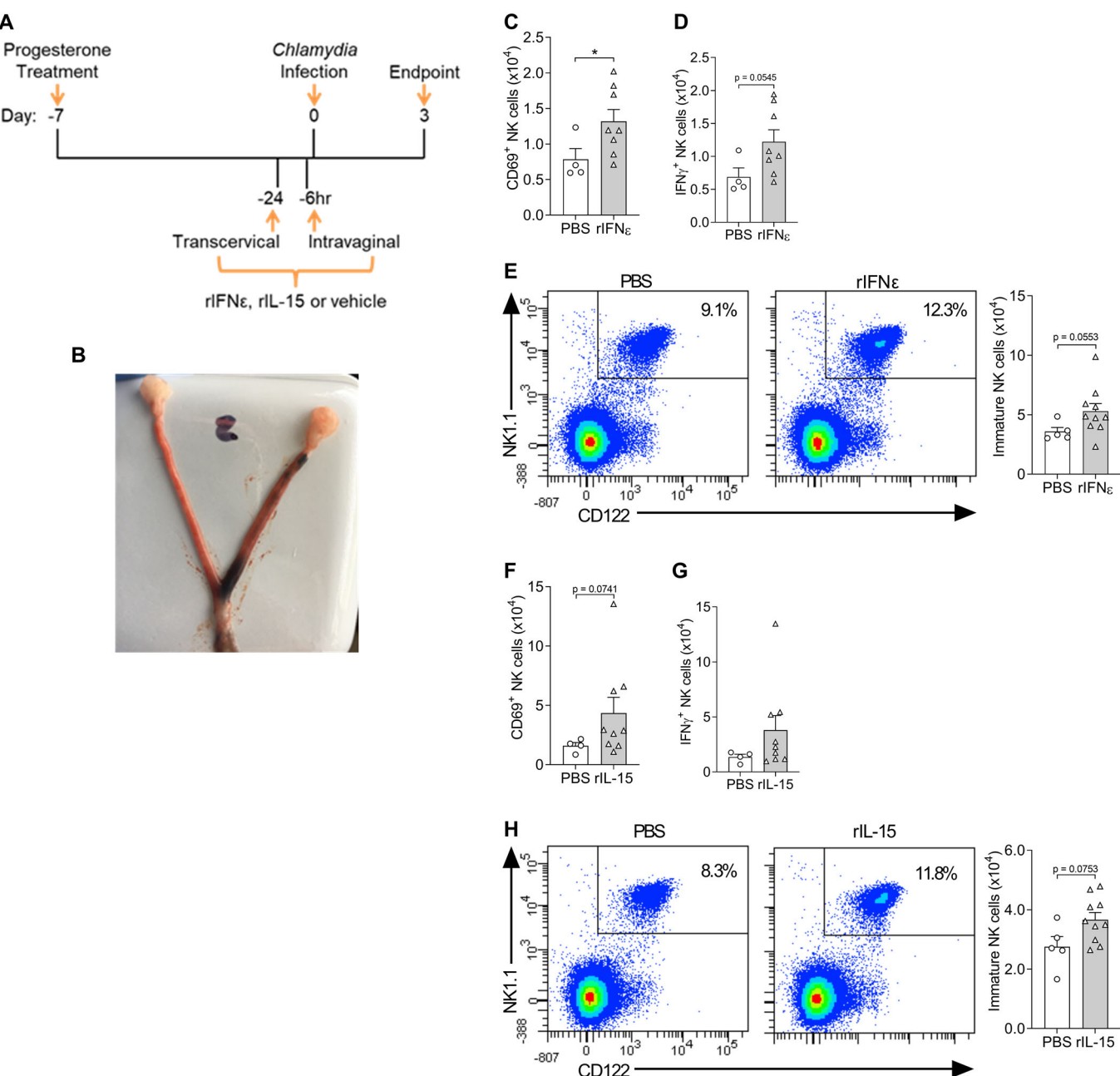

**Figure EV4.  In vivo administration and effect of recombinant (r)interferon (IFN)ε or rIL-15 on the numbers of activated and IFNγ-producing NK cells in the uterus and immature NK cells in bone marrow during *Chlamydia* infection.**

(A) Seven days prior to infection mice were administered progesterone subcutaneously. Twenty-four hours prior to infection rIFNε, rIL-15 or phosphate-buffered saline (PBS))/PBS with 0.1% bovine serum albumin (BSA) vehicle was delivered into the uterine lumen using a small animal endoscope. Six hours prior to infection rIFNε, rIL-15 or vehicle was delivered into the vagina using a pipette. On day 0 mice were infected intravaginally with *C. muridarum*. At 3 days post infection, mice were culled and immune responses assessed. (B) Staining throughout uterus of mouse administered 100 μL Evan's blue dye transcervically demonstrating the distribution of substances delivered using this technique. (C, D) Quantification of (C) CD69+ and (D) IFNγ+ conventional NK cells (FSC^low-int^ SSC^low^ CD45+ CD3- NK1.1+) in uterine horns from wild-type (WT) C57BL/6 mice prophylactically administered rIFNε or phosphate-buffered saline (PBS) vehicle control transcervically on day 3 of *Chlamydia muridarum* infection. (E) Flow cytometry of bone marrow from femurs as in A, showing immature conventional NK cells (FSC^low-int^ SSC^low^ CD45+ lin- CD11b- CD122+ NK1.1+) and quantification. (F, G) Quantification of (F) CD69+ and (G) IFNγ+ conventional NK cells (FSC^low-int^ SSC^low^ CD45+ CD3- NK1.1+) in uterine horns from wild-type (WT) C57BL/6 mice prophylactically administered rIL-15 or 0.1% bovine serum albumin (BSA) in phosphate-buffered saline (PBS) vehicle control transcervically on day 3 of *Chlamydia muridarum* infection. (H) Flow cytometry of bone marrow from femurs as in (D), showing immature conventional NK cells (FSC^low-int^ SSC^low^ CD45+ lin- CD11b- CD122+ NK1.1+) and quantification. Data information: The % displayed on the flow cytometry plots are the % of the parent population the cells within the gates comprise. All data presented as mean ± SEM, with individual values. *$p < 0.05$ ((A, B, D, E): one-tailed Mann–Whitney test; (C, F): two-tailed Mann–Whitney test). (C–E): $n \geq 4$ (data from one experiment), (F–H): $n \geq 4$ (data from one experiment; all biological replicates).

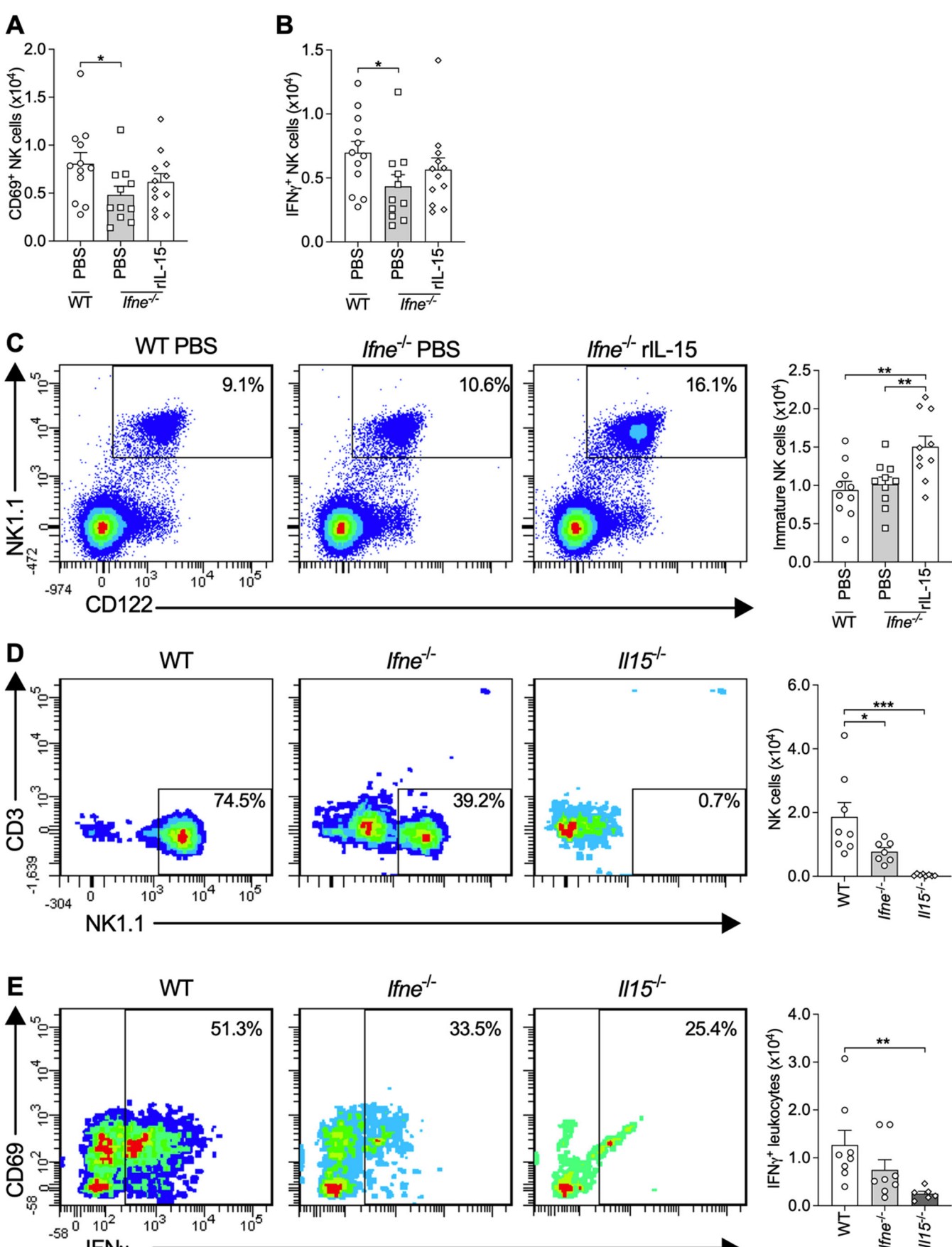

◀ **Figure EV5. Recombinant (r)IL-15 has no effect on the numbers of activated and interferon (IFN)γ-producing NK cells in the uterus and increases immature NK cells in the bone marrow in *Ifne*-deficient (-/-) mice and *Il15*-/- mice have no NK cells and reduced IFNγ+ leukocytes in the uterus during *Chlamydia* infection.**

(A, B) Quantification of (A) CD69+ and (B) IFNγ+ conventional NK cells (FSC$^{low-int}$ SSC$^{low}$ CD45+ CD3- NK1.1+) in uterine horns from *Ifne*-/- and wild-type (WT) C57BL/6 mice prophylactically administered rIL-15 (*Ifne*-/-) or 0.1% bovine serum albumin (BSA) in phosphate-buffered saline (PBS) vehicle control (*Ifne*-/- and WT) transcervically on day 3 of *Chlamydia muridarum* infection. (C) Flow cytometry of bone marrow from femurs as in (A) showing immature conventional NK cells (FSC$^{low-int}$ SSC$^{low}$ CD45+ lin- CD11b- CD122+ NK1.1+) and quantification. (D) Flow cytometry of uterine horn cells from *Il15*-/-, *Ifne*-/- and wild-type (WT) C57BL/6 mice on day 3 of *Chlamydia muridarum* infection showing conventional NK cells (FSC$^{low-int}$ SSC$^{low}$ CD45+ CD3- NK1.1+) and quantification. (E) Flow cytometry of uterine horn cells, showing IFNγ+ leukocytes (CD45+ cells) and quantification. Data information: The % displayed on the flow cytometry plots are the % of the parent population the cells within the gates comprise. All data presented as mean ± SEM, with individual values. *$p < 0.05$, **$p < 0.01$, ***$p < 0.001$ (one-way ANOVA). (A, B): $n \geq 11$, (C): $n = 10$ (data from one experiment), (D, E): $n = 8$ (data from one experiment; all biological replicates).

