## [Peer Review File · EMBO Molecular Medicine]

Interferon-epsilon is a novel regulator of NK cell responses in the uterus

Philip Hansbro, Jemma Mayall, Niamh Mangan, Anne Chevalier, Huw McCarthy, Daniel Hampsey, Chantal Donovan, Alexandra Brown, Anthony Matthews, Nicole de Weerd, Malcolm Starkey, Richard Kim, Bridie Goggins, Simon Keely, Steven Maltby, Rannay Baldwin, Paul Foster, Michael Boyle, Pradeep Tanwar, Nicholas Huntington, Paul Hertzog, Eveline de Geus, Katie Daly, and Jay Horvat

DOI: [10.15252/emmm.202115327](https://doi.org/10.15252/emmm.202115327)

Corresponding author: Philip Hansbro (Philip.Hansbro@newcastle.edu.au)

Review Timeline:

Submission Date:	22nd Oct 21
Editorial Decision:	15th Nov 21
Revision Received:	10th Aug 23
Editorial Decision:	14th Sep 23
Revision Received:	13th Dec 23
Accepted:	13th Dec 23

Editor: Zeljko Durdevic

Transaction Report:

15th Nov 2021

Dear Prof. Hansbro,

Thank you for the submission of your manuscript to EMBO Molecular Medicine. We have now received feedback from the three reviewers who agreed to evaluate your manuscript. As you will see from the reports below, the referees acknowledge the interest of the study but also raise important critique that should be addressed in a major revision. The concern about the *C. muridarum* murine infection model raised by the referee #2 should be addressed by discussing advantages but also limitations of the model for your study. You should also cite relevant publications that documented the appropriateness of the model to study *Chlamydia* infections of the female reproductive tract.

We would welcome the submission of a revised version within three months for further consideration. Please let us know if you require longer to complete the revision.

I look forward to receiving your revised manuscript.

Yours sincerely,

Zeljko Durdevic

We require:

- 1) A .docx formatted version of the manuscript text (including legends for main figures, EV figures and tables). Please make sure that the changes are highlighted to be clearly visible.
- 2) Individual production quality figure files as .eps, .tif, .jpg (one file per figure). For guidance, download the 'Figure Guide PDF': (<https://www.embopress.org/page/journal/17574684/authorguide#figureformat>).
- 3) A .docx formatted letter INCLUDING the reviewers' reports and your detailed point-by-point responses to their comments. As part of the EMBO Press transparent editorial process, the point-by-point response is part of the Review Process File (RPF), which will be published alongside your paper.
- 4) A complete author checklist, which you can download from our author guidelines (<https://www.embopress.org/page/journal/17574684/authorguide#submissionofrevisions>). Please insert information in the checklist that is also reflected in the manuscript. The completed author checklist will also be part of the RPF.
- 5) Please note that all corresponding authors are required to supply an ORCID ID for their name upon submission of a revised manuscript.
- 6) It is mandatory to include a 'Data Availability' section after the Materials and Methods. Before submitting your revision, primary

datasets produced in this study need to be deposited in an appropriate public database, and the accession numbers and database listed under 'Data Availability'. Please remember to provide a reviewer password if the datasets are not yet public (see <https://www.embopress.org/page/journal/17574684/authorguide#dataavailability>).

8) We would also encourage you to include the source data for figure panels that show essential data. Numerical data should be provided as individual .xls or .csv files (including a tab describing the data). For blots or microscopy, uncropped images should be submitted (using a zip archive if multiple images need to be supplied for one panel). Additional information on source data and instruction on how to label the files are available at

9) Our journal encourages inclusion of *data citations in the reference list* to directly cite datasets that were re-used and obtained from public databases. Data citations in the article text are distinct from normal bibliographical citations and should directly link to the database records from which the data can be accessed. In the main text, data citations are formatted as follows: "Data ref: Smith et al, 2001" or "Data ref: NCBI Sequence Read Archive PRJNA342805, 2017". In the Reference list, data citations must be labeled with "[DATASET]". A data reference must provide the database name, accession number/identifiers and a resolvable link to the landing page from which the data can be accessed at the end of the reference. Further instructions are available at

10) We replaced Supplementary Information with Expanded View (EV) Figures and Tables that are collapsible/expandable online. A maximum of 5 EV Figures can be typeset. EV Figures should be cited as "Figure EV1, Figure EV2" etc... in the text and their respective legends should be included in the main text after the legends of regular figures.

13) Author contributions: the contribution of every author must be detailed in a separate section (before the acknowledgments).

14) A Conflict of Interest statement should be provided in the main text.

***** Reviewer's comments *****

Referee #1 (Remarks for Author):

This manuscript extends on the previous finding of the groups of the senior authors on the role of interferon ϵ (IFN ϵ) in the female reproductive tract (FRT). Here they show that IFN ϵ plays a role in promoting the infiltration, activation, and IFN γ production of NK cells in the FRT during Chlamydia infection. This protection requires IL-15 which is induced by IFN ϵ , and by promoting the production of NK cell progenitor populations.

There is currently very little information on the immune response in the FRT as a consequence of epithelial infection by Chlamydia. This is a well-controlled study which provides convincing evidence that IFN ϵ is required to orchestrate the NK cell response against Chlamydia infection. Although the recruitment of NK cells to the infected tissue (e.g. Barth et al., Inf. Immun., 2021) and their pot. role in infection control has been published before, this study goes far beyond the current knowledge and provides mechanistic insight.

There are only few points that should be considered:

1. Earlier work by Tseng and Rank (PMID: 9826367) has demonstrated that depletion of NK cells significantly prolongs Chlamydia infection beyond 20 days. The current work shows now that infections are higher in the Ifne $^{-/-}$ mouse at 3 days but how is this in long-term infections? How much of the IFN ϵ infection control defect is due to a NK cell defect?
2. Since *C. muridarum* infections cause severe pathology in the upper female mouse genital tract, it would be important to link the IFN ϵ induced immune response to this clinically relevant outcome. How is IFN ϵ -dependent NK cell activation linked to this pathology?

Minor:

Figure 3F: The glandular IFN ϵ signal and the double positive cells should be additionally indicated with different arrows.

Page 7, 3rd paragraph: "... detected in uterine stroma (Figure 4; arrows)." This should be Fig. 3.

Referee #2 (Comments on Novelty/Model System for Author):

There are numerous anatomical differences between the murine and human female reproductive tract, and the site of infection by *C. trachomatis* also differs. In humans, the primary site of infection is the endocervix (which is a single cell-layer thick), whereas in the mouse model used here, it is the vagina. The mechanism by which IFN γ acts to restrict chlamydial growth differs dramatically between murine and human epithelial cells. Finally, there are vast genetic differences between *C. muridarum* and *C. trachomatis* - especially in how they respond to the effects of host-produced IFN γ .

These reasons sharply limit the translational potential of these studies conducted using *C. muridarum* to vaginally infect Depo-Provera treated mice to *C. trachomatis* infections in humans.

Referee #2 (Remarks for Author):

Mayall et al have evaluated the role of IFN ϵ as a regulator of NK cell responses in the uterus using a *C. muridarum* murine infection model. The primary findings in this manuscript are:

- 1) Expression of IFN ϵ in the uterus (largely/entirely by epithelial cells) is critical to the recruitment of NK cells to uterine tissue as well as their activation to produce IFN γ .
- 2) Ifne $^{-/-}$ mice have a modest decrease in the number of uterine pre-pro NK cells, but do not have a systemic decrease in the bone marrow, spleen, and lymph nodes.

3) IFN ϵ acts by stimulating CD45⁺ local immune cells to produce IL-15. Further, IL-15 and IFN ϵ are apparently produced by different subsets of CD11b⁺ cells.

4) IFN ϵ and IL-15 synergistically increase the proportion of NK cells that can produce IFN γ .

5) At least in vitro, treatment with recombinant IL-15 increases IFN γ production and the cytolytic activity of NK cells.

In general, the data are of high quality, and support the authors' interpretations. However, several issues remain to be clarified, including:

1) What are the cells that produce IL-15? Are they circulating monocytes that home to mucosal tissue in an IFN ϵ -dependent manner? Or are they some other type of ILC or dendritic cell?

2) While decreases in the relative expression of IL-15 are observed in Ifn ϵ ^{-/-} mice, does this reflect a smaller number of IL-15 producing cells recruited to uterine tissue v/s lower production of IL-15 per cell?

3) Along these lines, when mice are administered rIFN ϵ , is there an increase in the number of IL-15 producing cells recruited to uterine tissue? At least in humans, HPRT expression can vary quite a bit between tissues and cell-types. If this is also true in mice, it is difficult to use hprt-normalization as a method to normalize for total cell numbers.

4) Given the strong effect of IL-15 on IFN γ production by NK-cells, why is the effect of rIL-15 on reducing *C. muridarum* burden so modest (less than one log)?

What bearing do these findings in the murine FRT have for *C. trachomatis* infection in the human FRT, where the primary site of infection is a single cell-layer thick?

Referee #3 (Comments on Novelty/Model System for Author):

The animal model is adequate and the authors included examination of human uterine tissues for relationship of NK cells to IFN expression.

Referee #3 (Remarks for Author):

Very nice paper describing novel effects of IFN- ϵ in the uterus driving NK cell responses - including being essential for normal NK cell infiltration, NK cell activation and IFN γ production after Chlamydia infection. Detailed experiments examine the role of IFN ϵ -induced myeloid cell production of IL-15 in driving NK cell infiltration and activation. The use of rIL-15 locally revealed its stimulatory effects on NK cells but no increase in production of IFN ϵ , revealing that IL15 acts downstream of IFN ϵ . IL15^{-/-} mice were examined and revealed the importance of IL15 on NK cell development - IL15^{-/-} mice had no mature NK cells - these mice had lower levels of IFN γ -producing cells, IFN γ levels in the uterus, and higher chlamydial burdens. IFN ϵ treatment fails to induce IFN γ responses in IL15^{-/-} mice, again demonstrating IFN ϵ treatment acts through IL15 to stimulate NK cells to make IFN γ and that NK cells are essential for the protective effects against Chlamydia.

Adequate controls are included and the majority of the figures are easy to follow and detailed. Figure 3 F is impossible to see the fluorescent cells - a higher resolution image needs to be included.

It was somewhat disappointing that the investigators failed to examine the effects of IFN ϵ treatment in IL15-sufficient but NK cell deficient mice to determine if there would be any NK cell independent effects of IFN ϵ on early control of chlamydial infection via other resident tissue or early innate immune cell infiltrates.

The authors should also be careful to distinguish when they are reporting results of infected mice versus uninfected mice. At times it is confusing and it may be easier for the reader if results related to IFN ϵ deficiency in the absence of infection are presented first.

Point-by-point response EMM-2021-15327**Associate Editor**

We have now received feedback from the three reviewers who agreed to evaluate your manuscript. As you will see from the reports below, the referees acknowledge the interest of the study but also raise important critique that should be addressed in a major revision. The concern about the *C. muridarum* murine infection model raised by the referee #2 should be addressed by discussing advantages but also limitations of the model for your study. You should also cite relevant publications that documented the appropriateness of the model to study Chlamydia infections of the female reproductive tract.

Response: We thank the Editors and Reviewers for their interest and positive appraisal. We have addressed all of the Reviewers' comments in detail. We have addressed the comments on the *C. muridarum* murine infection model discussing its advantages and limitations, including references to its appropriateness for studying chlamydial infections of the female reproductive tract.

Reviewer 1

Comment (C)1: This manuscript extends on the previous finding of the groups of the senior authors on the role of interferon ϵ (IFN ϵ) in the female reproductive tract (FRT). Here they show that IFN ϵ plays a role in promoting the infiltration, activation, and IFN γ production of NK cells in the FRT during Chlamydia infection. This protection requires IL-15 which is induced by IFN ϵ , and by promoting the production of NK cell progenitor populations. There is currently very little information on the immune response in the FRT as a consequence of epithelial infection by *Chlamydia*. This is a well-controlled study which provides convincing evidence that IFN ϵ is required to orchestrate the NK cell response against Chlamydia infection. Although the recruitment of NK cells to the infected tissue (e.g. Barth et al., Inf. Immun., 2021) and their pot. role in infection control has been published before, this study goes far beyond the current knowledge and provides mechanistic insight.

Response (R)1: We thank the reviewer for their overwhelmingly positive appraisal of our manuscript, its quality and contribution.

C2: Earlier work by Tseng and Rank (PMID: 9826367) has demonstrated that depletion of NK cells significantly prolongs *Chlamydia* infection beyond 20 days. The current work shows now that infections are higher in the *Ifne*^{-/-} mouse at 3 days but how is this in long-term infections? How much of the IFN ϵ infection control defect is due to a NK cell defect?

Response R2: We previously reported in *Science* that bacterial burden is higher in *Ifne*^{-/-} mice throughout infection, from as early as 3 days and out to 30 days, and that clearance of infection is delayed (PMID: 23449591). Since this impaired control was evident from the very early stages of infection, here we focused on identifying the innate responses responsible for this IFN ϵ -mediated protection. We assessed all the different major cell types in infection in WT and *Ifne*^{-/-} mice by flow cytometry (Figure Expanded View 1A). By far the major changes that we observed were in NK cell numbers and responses during the early stages of infection in *Ifne*^{-/-} mice (Figure EV1A). Thus, we focussed the current study on the role of IFN ϵ in regulating this key innate cell in the FRT. Tseng and Rank showed that depletion of NK cells is associated with delayed clearance of infection (PMID: 9826367) and here we show that IFN ϵ deficiency leads to reduction in NK cells that is associated with increased infection. However, we also recently showed that IFN ϵ has direct effects on infection in epithelial cells (PMID: 29187603). To assess this further, we performed studies in *Il15*^{-/-} mice, which are completely devoid of mature NK cells. We show that these mice have higher bacterial burden at 3 days and recombinant IFN ϵ treatment in these mice is unable to increase IFN γ expression or decrease infection levels (Figure 7J, K, L). We recognise that we cannot fully delineate the effects of IFN ϵ -mediated NK cell responses versus those of other cells in controlling the level of infection. However, in combination, our data provide strong evidence that IFN ϵ plays an important role in protecting the FRT from infection through both suppressing infection in epithelial cells and increasing NK cell responses. Our findings shown here and those of Tseng and Rank show that IFN ϵ mediates its protective effects largely through promoting NK cell responses. We have now included the following statement to the Discussion (page 11):

“Here, we demonstrate that IFN ϵ deficiency leads to a reduction in NK cell numbers that is associated with increased infection and upper RT pathology. We also used NK cell deficient *Il15*^{-/-} mice and rIFN ϵ to show that NK cells are the major contributors to IFN ϵ -mediated infection control *in vivo*. This extends our previous studies where we found that bacterial burden is higher throughout *Chlamydia* infection in *Ifne*^{-/-} mice, from as early as 3dpi and out to 30dpi (Fung *et al.*, 2013). Notably, Tseng and Rank, showed that depletion of NK cells

is associated with delayed clearance of *Chlamydia* infection, however, their study did not investigate the role of NK cells in protection against ascending infection and pathology in the upper FRT (Tseng & Rank, 1998). We also previously found that IFN ϵ has direct effects on infection in epithelial cells (Stifter *et al.*, 2018). Thus, collectively, these findings show that IFN ϵ has important roles in protecting the FRT from infection and associated pathology largely by increasing NK cell responses, in addition to directly suppressing infection in epithelial cells.”

C3: Since *C. muridarum* infections cause severe pathology in the upper female mouse genital tract, it would be important to link the IFN ϵ induced immune response to this clinically relevant outcome. How is IFN ϵ - dependent NK cell activation linked to this pathology?

R3: The focus of this study was on how IFN ϵ affects NK cell responses as part of the innate host response to *Chlamydia* FRT infection. Only one other study has shown that NK cell depletion results in increased susceptibility to *Chlamydia* infection (delay in clearance [PMID: 9826367]) and this did not investigate the effects of NK cells on ascending infection in the upper FRT tissues. However, we have now conducted a new study assessing long term effects, and at 14dpi we find a 25% increase in gross inflammation of the oviducts in *Ifne*^{-/-} mice. We have now added these data to the expanded view (new Figure EV1B) and the following text to the Results (page 4):

“These changes in NK cell numbers early during infection are associated with an increase in gross oviduct pathology, measured by oviduct cross-sectional area, in *Ifne*^{-/-} mice at 14dpi (Figure EV1B). Increases in oviduct size are indicative of the development of hydrosalpinx, one of the key features of *Chlamydia*-induced pathology in both humans and mice (Lee *et al.*, 2020).”

Together, our findings suggest that IFN ϵ plays an important role in protecting the FRT from both infection and associated pathology by increasing NK cell responses. Determining the relative contribution of IFN ϵ -mediated NK responses to this protection against long-term sequelae is beyond the scope of the current study which is focussed on how IFN ϵ regulates NK cell responses during the innate host response to infection in the FRT. We state in the Discussion (page 12):

“Thus, collectively, these findings show that IFN ϵ has important roles in protecting the FRT from infection and associated pathology, largely by increasing NK cell responses, in addition to directly suppressing infection in epithelial cells.”

And have added the following (page 16):

“We also have not determined the relative contribution of NK cell responses in long-term sequelae. Such studies would require a substantial optimisation and additional animal experimentation. These are beyond the scope of the current study that shows a novel functional relationship between IFN ϵ , IL-15 and NK cells in FRT host defence for the first time.”

C4: Figure 3F: The glandular IFN ϵ signal and the double positive cells should be additionally indicated with different arrows.

R4: To address this we have now changed the markers so that they are distinct for the IFN ϵ signal in epithelial cells versus IFN ϵ IL-15 double positive cells in the stroma.

C5: Page 7, 3rd paragraph: "... detected in uterine stroma (Figure 4; arrows)." This should be Fig. 3.

R5: Thank you. We have corrected this error.

Reviewer 2

C1: There are numerous anatomical differences between the murine and human female reproductive tract, and the site of infection by *C. trachomatis* also differs. In humans, the primary site of infection is the endocervix (which is a single cell-layer thick), whereas in the mouse model used here, it is the vagina. The mechanism by which IFN gamma acts to restrict chlamydial growth differs dramatically between murine and human epithelial cells. Finally, there are vast genetic differences between *C. muridarum* and *C. trachomatis* - especially in how they respond to the effects of host-produced IFN gamma. These reasons sharply limit the translational potential of these studies conducted using *C. muridarum* to vaginally infect Depo-Provera treated mice to *C. trachomatis* infections in humans.

R1: All the major advances in understanding the mechanisms of pathogenesis of *Chlamydia* reproductive tract infections have been made with studies conducted using mouse models with *Chlamydia muridarum* or

Chlamydia trachomatis serovar D. To address this the following has been included in the Discussion (page 15). Such a discussion of the advantages and limitations of the model necessitates the addition of a large amount of text that results in the revised manuscript exceeding the word limit.

“This study used *C. muridarum* in mice to define the relationships between IFN ϵ and NK cell responses in the uterus. The advantage of using *C. muridarum* in murine models is that it is the natural mouse pathogen making it the most appropriate serovar for investigating host-pathogen relationships in the FRT. This is especially true for the study of acute infection, protective immunity and pathogenesis (De Clercq *et al.*, 2013; Morrison & Caldwell, 2002). In mice a low inoculum of *C. muridarum* establishes a productive infection that ascends the FRT and induces sustained endometrial inflammation and tubal pathology in immune-sufficient mice, and therefore, it reliably recapitulates the course of *C. trachomatis* FRT infections observed in women. *C. trachomatis* is not a natural pathogen in mice, and in immune-sufficient animals, it does not ascend the FRT beyond the cervix. This is unless the cervix is bypassed by infecting via the intrauterine or intrabursal route (Gondek *et al.*, 2012; Tuffrey *et al.*, 1990) or extremely large inocula are administered ($\sim 10^7$ IFU) (Darville *et al.*, 1997). This bacterium also does not induce the cardinal features of *Chlamydia*-induced disease in women (Darville *et al.*, 1997; Gondek *et al.*, 2012). Furthermore, IFN ϵ is expressed by epithelial cells throughout the FRT but most strongly in the uterine epithelium, therefore, for these studies we used *C. muridarum* that reliably ascends into the epithelium of the upper genital tract to investigate the regulation of immune responses by IFN ϵ .

IFN γ is associated with protection in both *C. muridarum* and *C. trachomatis* infections, however, *Chlamydia* species do differ in their susceptibility to IFN γ -mediated immune responses, with *C. muridarum* more resistant to IFN γ -mediated effects (Morrison, 2000; Perry *et al.*, 1999). This is a limitation of our study, as IFN ϵ -mediated NK cell IFN γ responses may have a greater impact on *C. trachomatis* infection and suggests that *C. muridarum* and *C. trachomatis* could also differ in their resistance to other cytokines. Future studies are needed to determine if the novel IFN ϵ -mediated immune responses we describe during *C. muridarum* infection have a similar impact on *C. trachomatis*, and other *Chlamydia* spp. and FRT pathogens. Such studies would require substantial optimisation and additional animal experimentation. These are beyond the scope of the current study that shows a novel functional relationship between IFN ϵ , IL-15 and NK cells in FRT host defence for the first time.”

Conducting studies that examine the effects of IFN ϵ (or the absence of IFN ϵ) on NK cell responses during *C. trachomatis* infections in whole human endometrial tissue were not feasible. This was due to ethical and logistical difficulties in obtaining appropriate tissues at early time points during infection, and because studies in human tissues *ex vivo* do not incorporate the intense influx of immune cells from the circulation or the effects on bone marrow haematopoiesis, both of which are key responses to infection. To provide clinical relevance to our findings, we performed correlation analyses on endometrial tissue biopsies. This data that was in the original submission shows that, in women, IFN ϵ expression is linked to both the expression of IL-15 and NCR1, a marker for NK cell recruitment (Figure 3 C-E). To address this, we have added the following statement to the Discussion (page 16):

“To add further clinical relevance to our findings we performed correlation analysis on endometrial biopsies. We show that in women, IFN ϵ expression is linked to both the expression of IL-15 and NCR1, a marker of NK cell recruitment. It was not possible to obtain endometrial tissues at early time points during infection due to ethical and logistical reasons, and studies in human tissues *ex vivo* do not incorporate the influx of immune cells from the circulation or the effects on bone marrow haematopoiesis, which are key responses to infection.”

C2: What are the cells that produce IL-15? Are they circulating monocytes that home to mucosal tissue in an IFN epsilon-dependent manner? Or are they some other type of ILC or dendritic cell? While decreases in the relative expression of IL-15 are observed in *Ifne*^{-/-} mice, does this reflect a smaller number of IL-15 producing cells recruited to uterine tissue v/s lower production of IL-15 per cell?

R2: Our data shows that the constitutive expression of IFN ϵ drives IL-15 expression in WT mice, as *Ifne*^{-/-} mice have reduced IL-15 gene and protein expression in the uterus both at baseline and during infection (Figure 3A,B). To further address this we have now performed additional analyses to identify the source of IFN ϵ -mediated IL-15 production. We now show that IL-15 producing cells in uterine tissue are predominantly (95.6%) CD11b⁺ CD11c⁻ cells. Of these, 89% are Ly6G⁻ MHC-II⁻, indicating that they are likely monocytes/macrophages (new Figures 3G-L). In initial profiling experiments (Figure EV1A), we show that the numbers of macrophages, mDCs and pDCs in the uterus are unchanged between WT and *Ifne*^{-/-} mice. Together,

these data suggest that the constitutive expression of IFN ϵ drives the increased production of IL-15 by monocytes/macrophages but not their infiltration into the uterus. We recognise that performing gain (i.e. rIFN ϵ) and loss (IFN ϵ KO) of IFN ϵ function studies in IL-15 reporter mice is required. This would fully identify which cells produce IL-15 in response to IFN ϵ and whether IFN ϵ induces the recruitment of IL-15-producing cells versus an increase in the production of IL-15 per cell. However, the major focus of this study was to identify the role of IFN ϵ in FRT immunity as opposed to identifying the cellular source of IL-15 in the uterus. Additionally, re-deriving IL-15 reporter mice and either administering rIFN ϵ or backcrossing with *Ifne*^{-/-} mice are major undertakings and not feasible due to time and funding considerations. We have now edited Figure 3 and Figure EV3 to include this new data and make the identification of IL-15 *versus* IFN ϵ -producing cells clearer.

To address these issues in the manuscript we now include this new data defining IL-15 producing cells in the Results section (page 7-8):

“We then performed flow cytometry to further characterize these cells (Figure EV3D-G). As expected, almost all epithelial cells expressed IFN ϵ (Figure 3G), but not IL-15 (Figure 3H). Notably, a large population of CD45⁺ leukocytes expressed IL-15 (Figure 3H) while only a small subset of leukocytes expressed IFN ϵ (Figure 3G). Further analysis of the IL-15⁺CD45⁺ population revealed that these cells did not express NK cell (NKp46), B cell (CD19; Figure 3I), or T cell (CD4, CD8; Figure 3J) markers but were primarily CD11b⁺ CD11c^{neg-low} MHC-II⁻ Ly6G⁻ cells (Figure 3K-L), a surface marker expression pattern indicative of monocytes/macrophages. To determine the extent to which IL-15 is co-expressed with IFN ϵ , we then characterised the expression of both cytokines in common immune cell subsets in the uterus. In uninfected uteri, NK cells are a minor immune cell population (~1-2% of CD45⁺ cells). Small proportions of the immune cell subsets assessed co-expressed IFN ϵ and IL-15, with 14.2% of NK cells, 2% of B cells, 2% of CD4⁺ cells, 3.2% of CD8⁺ cells and 5.5% of monocytes/macrophages expressing both cytokines (Figure EV3I-K).”

We have also added the following to the Discussion (page 13):

“We also show that IL-15 is produced by myeloid cells in response to IFN ϵ signaling from endometrial epithelial cells. The combination of high CD11b, low CD11c and absence of MHC-II expression indicates that these IL-15-producing cells are not one of the common DC subtypes, while the absence of Ly6G precludes neutrophils (Liu et al, 2020; Merad et al, 2013; Yasuda et al, 2020). This surface marker expression profile is consistent with monocyte/macrophage phenotypes reported across many mouse tissues (Liu et al., 2020). IL-15 is produced by macrophages and stromal cells in the decidua (Gordon, 2021), however, studies characterising cytokine and surface marker expression profiles of these cells in the non-pregnant mouse uterus are few. One limitation of our current study is that we do not identify whether IFN ϵ -mediated increases in IL-15 production are due to increases in the recruitment and/or proliferation of IL-15 producing cells in the uterus or increases in the amount of IL-15 produced per cell. Further investigation is required to determine the role of IFN ϵ in mediating IL-15 responses by monocytes/macrophages in the non-pregnant uterus.”

C3: Along these lines, when mice are administered rIFN ϵ , is there an increase in the number of IL-15 producing cells recruited to uterine tissue? At least in humans, HPRT expression can vary quite a bit between tissues and cell-types. If this is also true in mice, it is difficult to use hpert-normalization as a method to normalize for total cell numbers.

R3: We recognise the potential weaknesses associated with interpreting relative expression of target genes to housekeeping genes. However, HPRT has excellent stability in uterine tissue from hormonally treated mice (M value = 0.05, well below recommended cut-off of 0.5; PMID: 23638092). Notably, in addition to qPCR data that shows a reduction in IL-15 expression relative to HPRT in *Ifne*^{-/-} compared to WT mice, we also show that there is a decrease in IL-15 protein in the uterine lavage in *Ifne*^{-/-} mice (Figure 3B). Thus, we are confident that the changes in IL-15 expression reported relative to HPRT translate to changes at the protein level. To address this issue in the manuscript we have added the following to the Methods (page 21):

“Crucial qPCR data (e.g. *Il15* expression in uterine tissue) was confirmed by assessing protein levels in uterine lavage fluid by ELISA (Figure 3B).”

C4: Given the strong effect of IL-15 on IFN γ production by NK-cells, why is the effect of rIL-15 on reducing *C. muridarum* burden so modest (less than one log)?

R4: rIL-15 treatment *in vivo* had a modest effect in increasing IFN γ producing cells and reducing *Chlamydia* burden compared to the effect of complete genetic ablation of IL-15 (Figure 7I & J) or IFN γ signalling (PMID:

9169744, PMID: 9038313, PMID: 10496942) on *Chlamydia* infection. This likely mirrors the degree of loss/gain of function of protective NK cell and IFN γ responses in these studies and rIL-15 not reaching the target cells and/or for a sufficient period of time. Higher doses of rIL-15 or alternate routes of administration may be required to induce stronger protective NK cell IFN γ responses *in vivo* and have a greater impact on *Chlamydia* burden. Additionally, IFN ϵ levels are not altered during rIL-15 treatment (Figure 6I), and we show in our *in vitro* studies (Figure 4) that co-stimulation with both cytokines is required to maximise NK cell responses. To address this in the manuscript we have added the following to the Discussion (page 14):

“rIL-15 treatment *in vivo* had a modest effect in reducing *Chlamydia* burden compared to the effect of complete genetic ablation of IL-15. This is likely due to the degree of loss/gain of function in these experiments and rIL-15 not reaching the target cells and/or for a sufficient duration. Higher doses of rIL-15 or alternate routes of administration may be required to induce stronger protective NK cell IFN γ responses *in vivo* and have a greater impact on *Chlamydia* burden.”

C5: What bearing do these findings in the murine FRT this have for *C. trachomatis* infection in the human FRT, where the primary site of infection is a single cell-layer thick?

R5: In the original submission we performed correlation analyses on available uterine biopsy tissues to translate our findings of the relationship between IFN ϵ , IL-15 and NK cells to the uteri of human subjects (Figure 3C-E). Our study focused on IFN ϵ -mediated immune responses in the uterus and the role of these in ascending infection as this is the primary site of IFN ϵ expression and where much of the *Chlamydia*-induced pathology occurs. Endometrial host cells in the upper FRT are histologically similar between humans and mice, consisting of simple columnar luminal epithelium and tubular glands within a specialised stroma (PMID: 33796844; DOI: 10.1016/B978-0-12-381361-9.00017-2). Studies investigating innate immune processes during FRT infection are difficult to perform in human subjects *in situ*. This is due to ethical and logistical considerations in obtaining appropriate tissues at appropriate time points during early infection. Furthermore, studies in human tissues *ex vivo* do not incorporate the intense influx of immune cells from the circulation or the effects on bone marrow haematopoiesis, both of which are key during infection. Our findings will spur future studies that will further investigate the relationship between IFN ϵ , NK cells, IFN γ responses and protection against infection throughout the entire FRT in women. Please see **R1** that describes how we have modified the manuscript to address this comment. We also added the following to the Discussion (page 16):

“Endometrial cells in the upper FRT are histologically similar between humans and mice, consisting of simple columnar luminal epithelium and tubular glands within a specialised stroma (Rendi *et al*, 2012; Yamaguchi *et al*, 2021).”

Reviewer 3

C1: Figure 3F is impossible to see the fluorescent cells - a higher resolution image needs to be included.

R1: Thank you for highlighting this. To address this we have now submitted the individual micrograph panels as the quality of the image seems to be affected when inserted into the manuscript's Word document.

C2: It was somewhat disappointing that the investigators failed to examine the effects of IFN ϵ treatment in IL15-sufficient but NK cell deficient mice to determine if there would be any NK cell independent effects of IFN ϵ on early control of *Chlamydia* infection *via* other resident tissue or early innate immune cell infiltrates.

R2: We did consider this, however, early during infection, IFN ϵ had by far its major effects on NK cells but little effect on all other cell types examined (Figure EV1A). To address this issue we have now also performed staining for innate lymphoid cell (ILC) populations (ILC1-3: CD45⁺ Lin⁻ IL-7R α ⁺ CD90.2⁺ T-bet^{+/-}) in the uterus, however, they are absent, or too few present to examine, during *Chlamydia* infection (data below).

This again shows that NK cells are the main contributors to IFN ϵ -mediated protective responses. We have already published on the direct effects of IFN ϵ on epithelial cells (PMID: 29187603). Here, we focus on defining the mechanisms of IFN ϵ -mediated NK cell responses. We did utilize IL-15-deficient mice, which are devoid of mature NK cells, to determine the NK cell-independent effects of IFN ϵ *in vivo*. We recognise that IL-15 deficiency will have effects on other cell types (*Il15*^{-/-} mice also exhibit defects in NKT, IEL, and memory CD8 T cells) and that these mice are not exclusively NK cell deficient (PMID: 24492800). However, all currently available tools for depleting NK cells have non-specific effects. There are no markers expressed exclusively by NK cells to target with antibody depletion that are not also expressed by other cell types. Genetic deletion of factors required for NK cell development and function commonly affect T and B cell function, lead to the development of severe autoimmune disorders, or only target cytolytic activity (PMID: 26237009). Thus, *Il15*^{-/-} mice are the most appropriate tool available to determine if there are NK cell-independent effects of IFN ϵ on infection and protective responses. We hope that tools for the investigation of NK cell biology continue to improve and allow for more targeted studies in the future. To address this comment we have added the following to the manuscript.

We have added the new ILC data (new Figure EV1B and associated text) with the following statement to the Results (page 4).

“We also assessed the effects on innate lymphoid cells (ILC1-3: CD45⁺ Lin⁻ IL-7R α ⁺ CD90.2⁺ T-bet^{+/+}) in the uterus, however, they were absent, or too few present to examine, during *Chlamydia* infection (Figure EV1B).”

We have added discussion of the use of NK deficient mice to the Discussion (page 16):

“It would also be informative to assess IL-15 sufficient but NK cell deficient mice to further assess any NK cell independent effects of IFN ϵ *via* other resident tissue or early innate immune cells. However, collectively our data show that NK cells are the main contributors to IFN ϵ -mediated protective responses. We have previously shown the direct effects of IFN ϵ on epithelial cells (Stifter *et al.*, 2018), and here we show that numbers of other common immune cell populations in the uterus are unaffected by IFN ϵ deficiency, and use *Il15*^{-/-} mice, that are devoid of mature NK cells, to determine the NK cell-independent effects of IFN ϵ *in vivo*. IL-15 deficiency may have effects on other cell types (*Il15*^{-/-} mice also exhibit defects in NKT, intraepithelial lymphocytes and memory CD8⁺ T cells) and so these mice are not exclusively NK cell deficient (Huntington, 2014). However, all currently available tools for depleting NK cells have non-specific effects and there are no markers expressed exclusively by NK cells to target with antibody depletion that are not also expressed by other cell types. Genetic deletion of factors required for NK cell development and function commonly affect T and B cell function, lead to the development of severe autoimmune disorders, or only target cytolytic activity (Zamora *et al.*, 2015). Thus, *Il15*^{-/-} mice are the most appropriate tool available to determine if there are NK cell-independent effects of IFN ϵ on infection and protective responses.”

C3: The authors should also be careful to distinguish when they are reporting results of infected mice versus uninfected mice. At times it is confusing and it may be easier for the reader if results related to IFN ϵ deficiency in the absence of infection are presented first.

R3: We thank the reviewer for this suggestion. We have gone through the manuscript and altered the text to enhance the clarity in reporting changes in uninfected versus infected mice. Where possible, results from uninfected mice are stated first, however, as there are few immune cells in the uterus of uninfected mice, much of our uterine tissue flow cytometry was performed on tissue from infected animals only.

14th Sep 2023

Dear Prof. Hansbro,

Thank you for the submission of your revised manuscript to EMBO Molecular Medicine, and please accept my apologies for the delay in getting back to you. We have now received feedback from 1 out of 2 reviewers whom we asked to re-evaluate your manuscript. As the referee #2 will unfortunately not be able to return his/her report in a timely manner we prefer to make a decision now in order to avoid further delay in the process. I am pleased to inform you that we will be able to accept your manuscript pending the following final amendments:

1) Author checklist: We updated our Author Checklist. Please submit a complete updated version.

<https://www.embopress.org/pb-assets/embo-site/EMBO%20Press%20Author%20Checklist-1642513524327.xlsx>

2) Figures:

- Remove all figures from the main manuscript file and leave only their legend.

- During a standard image analysis we detected possible reuse of images not stated in figure legends: Figure 1B in Figure EV1F, Figure 2A and E in Figure EV1 G, Figure 3 G and H in Figure EV3 F and G. Please check the composition of these figures yourself, and if the images/panels are reused please reference them in the figure legends.

3) Tables: Remove EV tables from the main manuscript file and upload them as separate files with their legends.

4) In the main manuscript file, please do the following:

- Correct/answer the track changes suggested by our data editors by working from the attached document.

- Add up to 5 keywords.

- In M&M, provide the antibody dilutions that were used for each antibody.

- In M&M, please include statement that the informed consent was obtained from all human subjects and that the experiments conformed to the principles set out in the WMA Declaration of Helsinki and the Department of Health and Human Services Belmont Report.

- Remove "List of supplementary material".

- Rename "Conflict of interest" to "Disclosure Statement & Competing Interests". We updated our journal's competing interests policy in January 2022 and request authors to consider both actual and perceived competing interests. Please review the policy <https://www.embopress.org/competing-interests> and update your competing interests if necessary.

- Author contributions: Please remove it from the manuscript and specify author contributions in our submission system. CRediT has replaced the traditional author contributions section because it offers a systematic machine-readable author contributions format that allows for more effective research assessment. You are encouraged to use the free text boxes beneath each contributing author's name to add specific details on the author's contribution. More information is available in our guide to authors:

<https://www.embopress.org/page/journal/17574684/authorguide#authorshipguidelines>

- In data availability section please replace the current sentence with following; This study includes no data deposited in external repositories.

5) Funding: Please make sure that information about all sources of funding are complete in both our submission system and in the manuscript. Currently, information about Hunter New England Local Health District (HNELHD) is missing in the manuscript.

6) The Paper Explained: Please provide "The Paper Explained" and add it to the main manuscript text. Please check "Author Guidelines" for more information. <https://www.embopress.org/page/journal/17574684/authorguide#researcharticleguide>

7) Synopsis:

- Synopsis image: Please provide a separate, high-resolution 550 px-wide x (250-400)-px high jpeg file.

- Please check your synopsis text and image and submit their final versions with your revised manuscript. Please be aware that in the proof stage minor corrections only are allowed (e.g., typos).

8) For more information: This space should be used to list relevant web links for further consultation by our readers. Could you identify some relevant ones and provide such information as well? Some examples are patient associations, relevant databases, OMIM/proteins/genes links, author's websites, etc...

9) Source data: We encourage you to include the source data for figure panels that show essential data. Numerical data should be provided as individual .xls or .csv files (including a tab describing the data). For blots or microscopy, uncropped images should be submitted (using a zip archive if multiple images need to be supplied for one panel). Please check "Author Guidelines" for more information. <https://www.embopress.org/page/journal/17574684/authorguide#sourcedata>

10) As part of the EMBO Publications transparent editorial process initiative (see our Editorial at <http://embomolmed.embopress.org/content/2/9/329>), EMBO Molecular Medicine will publish online a Review Process File (RPF) to accompany accepted manuscripts. This file will be published in conjunction with your paper and will include the anonymous referee reports, your point-by-point response and all pertinent correspondence relating to the manuscript. Let us know whether you agree with the publication of the RPF and as here, if you want to remove or not any figures from it prior to publication. Please note that the Authors checklist will be published at the end of the RPF.

11) Please provide a point-by-point letter INCLUDING my comments as well as the reviewer's reports and your detailed responses (as Word file).

I look forward to reading a new revised version of your manuscript as soon as possible.

Yours sincerely,

Zeljko Durdevic

*** Instructions to submit your revised manuscript ***

- 1) a .docx formatted version of the manuscript text (including Figure legends and tables)
- 2) Separate figure files*
- 3) supplemental information as Expanded View and/or Appendix. Please carefully check the authors guidelines for formatting Expanded view and Appendix figures and tables at <https://www.embopress.org/page/journal/17574684/authorguide#expandedview>
- 4) a letter INCLUDING the reviewer's reports and your detailed responses to their comments (as Word file).
- 5) The paper explained: EMBO Molecular Medicine articles are accompanied by a summary of the articles to emphasize the major findings in the paper and their medical implications for the non-specialist reader. Please provide a draft summary of your article highlighting
 - the medical issue you are addressing,
 - the results obtained and
 - their clinical impact.This may be edited to ensure that readers understand the significance and context of the research. Please refer to any of our published articles for an example.
- 6) For more information: There is space at the end of each article to list relevant web links for further consultation by our readers. Could you identify some relevant ones and provide such information as well? Some examples are patient associations, relevant databases, OMIM/proteins/genes links, author's websites, etc...
- 7) Author contributions: the contribution of every author must be detailed in a separate section.
- 8) EMBO Molecular Medicine now requires a complete author checklist (<https://www.embopress.org/page/journal/17574684/authorguide>) to be submitted with all revised manuscripts. Please use the checklist as guideline for the sort of information we need WITHIN the manuscript. The checklist should only be filled with page numbers where the information can be found. This is particularly important for animal reporting, antibody dilutions (missing) and exact values and n that should be indicated instead of a range.
- 9) Every published paper now includes a 'Synopsis' to further enhance discoverability. Synopses are displayed on the journal

webpage and are freely accessible to all readers. They include a short stand first (maximum of 300 characters, including space) as well as 2-5 one sentence bullet points that summarise the paper. Please write the bullet points to summarise the key NEW findings. They should be designed to be complementary to the abstract - i.e. not repeat the same text. We encourage inclusion of key acronyms and quantitative information (maximum of 30 words / bullet point). Please use the passive voice. Please attach these in a separate file or send them by email, we will incorporate them accordingly.

You are also welcome to suggest a striking image or visual abstract to illustrate your article. If you do please provide a jpeg file 550 px-wide x 400-px high.

10) A Conflict of Interest statement should be provided in the main text

11) Please note that we now mandate that all corresponding authors list an ORCID digital identifier. This takes <90 seconds to complete. We encourage all authors to supply an ORCID identifier, which will be linked to their name for unambiguous name identification.

Currently, our records indicate that the ORCID for your account is 0000-0002-4741-3035.

Link Not Available

12) The system will prompt you to fill in your funding and payment information. This will allow Wiley to send you a quote for the article processing charge (APC) in case of acceptance. This quote takes into account any reduction or fee waivers that you may be eligible for. Authors do not need to pay any fees before their manuscript is accepted and transferred to our publisher.

Photos 400-800 DPI

*Additional important information regarding figures and illustrations can be found at

<https://bit.ly/EMBOPressFigurePreparationGuideline>. See also figure legend preparation guidelines:

<https://www.embopress.org/page/journal/17574684/authorguide#figureformat>

The system will prompt you to fill in your funding and payment information. This will allow Wiley to send you a quote for the article processing charge (APC) in case of acceptance. This quote takes into account any reduction or fee waivers that you may be eligible for. Authors do not need to pay any fees before their manuscript is accepted and transferred to our publisher.

***** Reviewer's comments *****

Referee #3 (Comments on Novelty/Model System for Author):

All reviewers' critiques are well addressed. Study is of high quality.

Referee #3 (Remarks for Author):

All reviewers' critiques are well addressed. Study is of high quality and impactful with respect to increasing understanding of the mechanisms for IFNepsilon effects on chlamydial infection in FRT.

The authors addressed the minor formatting issues.

13th Dec 2023

Dear Prof. Hansbro,

Please find enclosed the final reports on your manuscript. We are pleased to inform you that your manuscript is accepted for publication and is now being sent to our publisher to be included in the next available issue of EMBO Molecular Medicine.
